# Dll1$^+$ quiescent tumor stem cells drive chemoresistance in breast cancer through NF-κB survival pathway

Sushil Kumar [1], Ajeya Nandi[1], Snahlata Singh[1], Rohan Regulapati [1], Ning Li [1], John W. Tobias[2], Christian W. Siebel[3], Mario Andres Blanco[1], Andres J. Klein-Szanto[4], Christopher Lengner [1], Alana L. Welm [5], Yibin Kang [6] & Rumela Chakrabarti[1✉]

Development of chemoresistance in breast cancer patients greatly increases mortality. Thus, understanding mechanisms underlying breast cancer resistance to chemotherapy is of paramount importance to overcome this clinical challenge. Although activated Notch receptors have been associated with chemoresistance in cancer, the specific Notch ligands and their molecular mechanisms leading to chemoresistance in breast cancer remain elusive. Using conditional knockout and reporter mouse models, we demonstrate that tumor cells expressing the Notch ligand Dll1 is important for tumor growth and metastasis and bear similarities to tumor-initiating cancer cells (TICs) in breast cancer. RNA-seq and ATAC-seq using reporter models and patient data demonstrated that NF-κB activation is downstream of Dll1 and is associated with a chemoresistant phenotype. Finally, pharmacological blocking of Dll1 or NF-κB pathway completely sensitizes Dll1$^+$ tumors to chemotherapy, highlighting therapeutic avenues for chemotherapy resistant breast cancer patients in the near future.

[1] Department of Biomedical Sciences, University of Pennsylvania, Philadelphia, PA 19104, USA. [2] Department of Cancer Biology, Perelman School of Medicine, University of Pennsylvania, Philadelphia, PA 19104, USA. [3] Department of Discovery Oncology, Genentech Inc., South San Francisco, CA 94080, USA. [4] Histopathology Facility, Fox Chase Cancer Center, Philadelphia, PA, USA. [5] Department of Oncological Sciences, Huntsman Cancer Institute, University of Utah, Salt Lake City, UT 84112, USA. [6] Department of Molecular Biology, Princeton University, Princeton, NJ 08544, USA. ✉email: rumela@vet.upenn.edu

Breast cancer is one of the leading causes of cancer-related death among women[1]. Despite recent progress in the field, clinical challenges including treatment resistance, recurrence, and metastasis still remain[2]. Breast cancer is classified into three main subtypes, luminal (expressing estrogen receptor (ER) with or without co-expression of the progesterone receptor (PR) and HER2 gene), HER2-enriched without ER and PR, and Triple-negative breast cancer (TNBC), which does not express any of these receptors[3,4]. Such intertumoral heterogeneity has so far precluded the identification of "universal" breast cancer therapeutics. Moreover, the known intratumoral heterogeneity of breast cancer is thought to directly contribute to chemotherapeutic resistance, as a small population of tumor cells known as tumor-initiating cells (TICs)/cancer stem cells (CSCs) often are endowed with quiescence, metastatic, and chemotherapy resistance capabilities[5–11] that promote tumor recurrence and metastasis, even after chemotherapeutic treatment[12,13].

TICs are present in several solid tumors including breast tumors[14], and Notch signaling has been specifically implicated in promoting TICs/CSC function in breast cancer[15–19]. The Notch pathway is composed of four Notch receptors (Notch1-4) and five ligands (Jagged1, Jagged2, Delta-like1 (Dll1), Dll3, and Dll4). While Notch signaling is dysregulated in breast cancer, and extensive studies support the function of Notch receptors in breast cancer[20–23], the role of specific Notch ligands in promoting TIC function and chemoresistance remains unclear. While global Notch signaling inhibitors reduce tumor growth by targeting TICs, their side effects limit its therapeutic efficacy. Notably, our recent studies highlighted the tumor-promoting function of the Notch ligand Dll1 in luminal breast cancer[24]. As specific ligand-based therapies are likely to provide a better safety profile than global Notch signaling inhibitors, determining the role of Dll1 in TIC generation/function and Notch-mediated chemotherapeutic resistance, may aid in the identification of effective and better-tolerated therapeutics for breast cancer patients.

In this study, we show that Dll1 is essential for tumor development and metastasis in an Polyoma Middle T (MMTV-PyMT) mammary tumor mouse model. Using reporter models, we further demonstrate a temporal increase of Dll1 expression in tumors as they progress from early to late stages, including metastasis. Mechanistically, we show that within luminal tumors, Dll1$^+$ cells represent quiescent TICs, display a chemotherapeutic resistant gene signature enriched in Hypoxia and NF-κB signaling genes, and are resilient to chemotherapy. Most notably, pharmacological blocking of Dll1 or NF-κB pathway sensitizes tumor cells to chemotherapy and significantly abolishes tumor growth and metastasis in tumor initiation and progression study. Collectively, these data suggest that quiescent Dll1$^+$ TICs are an important regulator of breast tumor progression and metastasis and are responsible for chemoresistance. As such, targeting Dll1 in combination with chemotherapy and NF-κB may be a promising therapeutic strategy for breast cancer patients who are resistant to chemotherapy.

## Results

### Dll1 conditional knockout reveals an oncogenic function for Dll1 in the luminal MMTV-PyMT but not in MMTV-Wnt1 model.

To dissect the role of Dll1 in autochthonous mammary tumors, we crossed *Dll1* conditional-knockout (*Dll1$^{cKO}$*) mice in which K14-Cre recombinase efficiently knocks out Dll1 in mammary epithelial cells[25,26] to MMTV-PyMT to generate PyMT-Dll1$^{cKO}$ animals (Fig. 1a). The MMTV-PyMT model represents a luminal-like breast cancer model with high metastatic potency[27,28], thereby permitting analysis of progression from early to late stages of the disease. Dll1 expression was

significantly downregulated in tumor cells of PyMT-Dll1$^{cKO}$ mice (Fig. 1b). Moreover, we found that loss of Dll1 led to a dramatic reduction in hyperplasia in PyMT-Dll1$^{cKO}$ compared to PyMT-Dll1$^{WT}$ mammary glands at 8 weeks (Fig. 1c), suggesting that loss of Dll1 affects early events in tumorigenesis, potentially by impacting tumor-initiating populations. In control PyMT-Dll1$^{WT}$ mice, mammary tumors occurred as early as 11 weeks of age, with almost 100% tumor penetrance. In contrast, tumors in PyMT-Dll1$^{cKO}$ mice occurred at 12 weeks of age with a 75% delayed penetrance (Fig. 1d). Moreover, PyMT-Dll1$^{WT}$ mice had a significantly increased tumor burden (Fig. 1e) and lung metastasis (Fig. 1f, g) compared to PyMT-Dll1$^{cKO}$ mice.

To determine if the function of Dll1 was specific to the luminal adenocarcinoma MMTV-PyMT model, we crossed the *Dll1$^{cKO}$* mice to MMTV-Wnt1 mice, which represents a basal/mixed tumor model[29], to generate Wnt1-Dll1$^{cKO}$ mice (Supplementary Fig. 1a). In contrast to the PyMT model, mammary glands from both *Wnt1-Dll1$^{cKO}$* and *Wnt1-Dll1$^{WT}$* mice exhibited similar hyperplastic growth (Supplementary Fig. 1b). Furthermore, there was no significant difference in tumor onset in *Wnt1-Dll1$^{cKO}$* compared with *Wnt1-Dll1$^{WT}$* mice ($p = 0.4$) (Supplementary Fig. 1c), suggesting a specific tumorigenic function for Dll1 in luminal adenocarcinoma.

### A dramatic increase of Dll1$^+$ tumor cells during PyMT tumor progression.

The dramatic effect of Dll1 deletion on PyMT tumor formation prompted us to investigate the spatial and temporal distribution of Dll1 expression during tumorigenesis. For this, we generated PyMT-*Dll1$^{mCherry}$* reporter mice[25,30] and evaluated the expression pattern of Dll1$^+$ cells during PyMT-Dll1$^{mCherry}$ tumor progression from normal mammary gland to late-stage tumor. We observed a substantial increase in the number of Dll1$^+$ cells as the mammary gland advanced through the hyperplastic stage to the tumor (Fig. 2a, b), supporting an oncogenic function Dll1 in PyMT tumors. The mRNA levels of Dll1 in sorted mCherry positive and negative populations from hyperplasia and tumors indicated faithful expression of the Dll1 in reporter mice (Fig. 2c). To analyze the status of Dll1$^+$ cells in basal and luminal cell compartments, we performed confocal microscopy using basal (K14 and K5) and luminal (K8) markers along with Dll1 mCherry staining. As previously reported, we found that Dll1 is predominantly expressed in the basal layer of the normal mammary gland[25] (Fig. 2d). However, there was a relative increase in the number of Dll1$^+$ luminal (K8$^+$) cells when compared to Dll1$^+$ basal tumor cells (K14$^+$ and K5$^+$) as tumors progressed from hyperplasia stage to late tumor (Fig. 2d, e and Supplementary Fig. 2a, b).

To further investigate the distribution of Dll1$^+$ cells during tumor development, we used PyMT-Dll1-GFP mouse model. We crossed Dll1$^{GFP-IRES-Cre-ERT2}$ knockin mice (Dll1$^{GFP}$)[31] with MMTV-PyMT mice to generate PyMT-Dll1$^{GFP}$ mice. The GFP expression faithfully recapitulated Dll1 expression (Supplementary Fig. 2c), and similar to PyMT-Dll1$^{mCherry}$ model, the number of Dll1$^+$ cells (GFP$^+$) in PyMT-Dll1$^{GFP}$ mice was significantly enhanced, and Dll1$^+$ cells acquired expression of the luminal marker (K8) as tumors progressed in this luminal model (Supplementary Fig. 2d–f and Supplementary Fig. 3a, b). These studies identified the cellular and temporal distribution of Dll1 at different stages of PyMT tumor development using multiple reporter models.

### Metastatic lung nodules show increased expression of Dll1 compared to primary tumors.

The MMTV-PyMT mouse model develops lung metastasis at a high frequency[28]. To determine if there is an association between the expression of Dll1 and

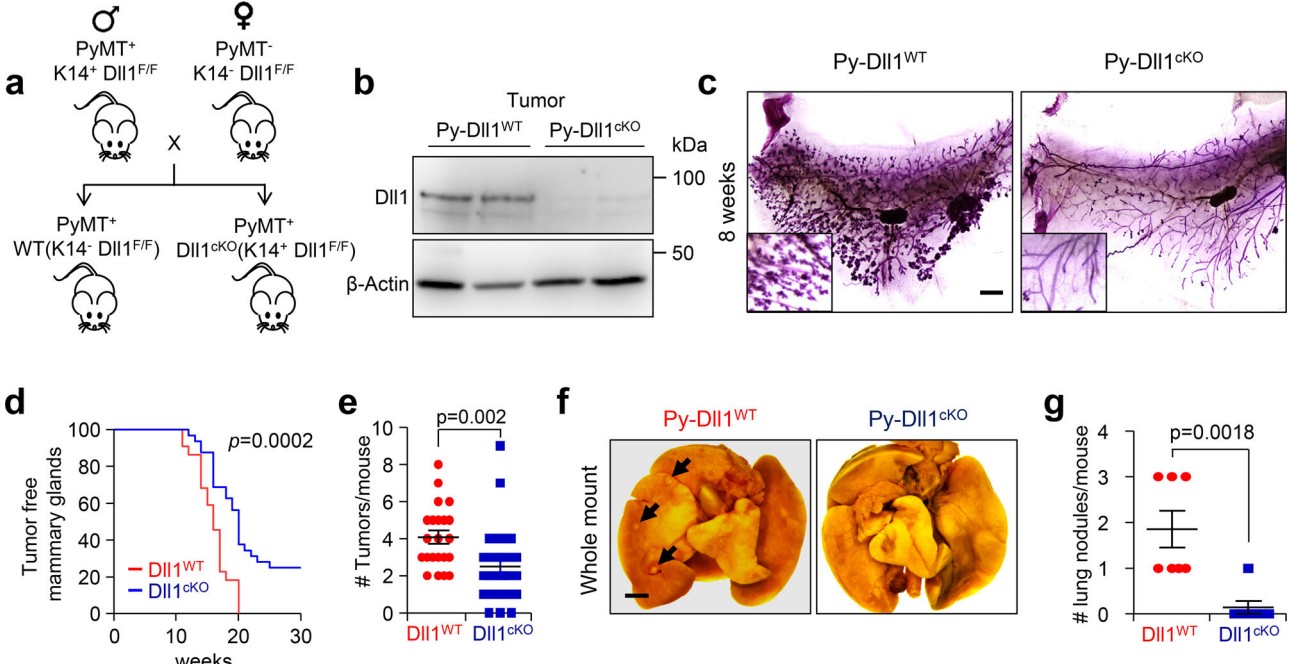

**Fig. 1 Conditional knockout of Dll1 significantly delays tumor initiation and lung metastasis in MMTV-PyMT model. a** Schematic shows the strategy to generate MMTV-PyMT; Dll1cKO mice. **b** Western blot shows reduced Dll1 protein in PyMT-Dll1cKO tumor cells compared to PyMT-Dll1WT tumor cells. **c** The representative whole mount alum carmine-stained images show the hyperplasia stages of MMTV-PyMT in indicated Dll1 genotypes. Insets show a zoomed-in image to a difference in branching. $n = 5$ PyMT-Dll1WT, $n = 6$ PyMT-Dll1cKO hyperplastic mammary glands at the age of 8 weeks. **d** Kinetics of mammary tumor onset in MMTV-PyMT females of indicated genotypes as shown by Kaplan–Meier plot. PyMT-Dll1WT ($n = 22$ mice) and PyMT-Dll1cKO ($n = 32$ mice) were used. **e** Scatter plot shows the number of spontaneous tumors in PyMT-Dll1WT ($n = 22$ mice) and PyMT-Dll1cKO ($n = 32$ mice). The mammary gland was considered tumor-positive when the tumor was $3 \times 3$ mm in dimension. **f** The representative whole mount images show metastatic lung nodules (black arrows) in PyMT-Dll1WT, which is absent in the PyMT-Dll1cKO mice. The lung nodules were quantified under dissection microscope after 24-h fixation with bouin's solution (please see method section for details). **g** The scatter plot shows the number of lung nodules in spontaneous tumors bearing PyMT-Dll1WT and PyMT-Dll1cKO mice ($n = 7$ mice per group). two-tailed Log-Rank test **d** and two-tailed unpaired Student's $t$-test **e, g** were used to calculate $p$ values. Data present two **b** or three **c** or seven **f** independent experiments. **e, g** Data are presented as the mean ± SEM. Scale bars, 500 µm **c** and 400 µm **f**. Source data are provided as a source data file.

metastasis, we assessed Dll1 expression in PyMT-Dll1mCh and PyMT-Dll1GFP lung metastatic nodules and compared them to primary tumors. We found higher levels of Dll1 expression in metastatic nodules in both reporter models as compared with the primary tumors (Supplementary Fig. 4a–f). Collectively, our studies from Dll1 conditional knockout (Fig. 1) and reporter mice suggest that Dll1 may be a critical determinant of metastasis in breast cancer.

**Dll1+ tumor cells display TIC characteristics**. Increased metastasis is often linked to the presence of Tumor-initiating cells (TICs)/Cancer stem cells (CSCs) in cancer[32–34]. To determine if Dll1+ cells display a TIC phenotype in breast tumors, we performed an in vitro tumorsphere assay to assess the tumorigenic potential of Dll1+ and Dll1− tumor cells isolated from PyMT-Dll1mCh tumors. Sorted single PyMT-Dll1+ tumor cells formed a significantly higher number of tumorspheres compared with PyMT-Dll1− tumor cells, suggesting PyMT-Dll1+ cells are enriched in TICs (Fig. 3a). To directly test the tumor-initiating function of Dll1, we transplanted sorted PyMT-Dll1+ or PyMT-Dll1− tumor cells into the mammary fat pad (MFP) of syngeneic C57BL/6 mice. We found that transplanted Dll1+ tumor cells initiated tumor growth before Dll1− tumor cells, suggesting that Dll1+ cells display characteristics of TICs (Fig. 3b). In limiting dilution assay, Dll1+ tumor cells showed a higher capability (1/ 2791) to initiate xenograft growth compared to Dll1− tumor cells (1/27,341) (Fig. 3c). Intriguingly, once Dll1− tumors started growing, they grew faster than Dll1+ tumors, suggesting a higher

proliferative potential of Dll1− tumor cells (Fig. 3d, e). As TICs have a slow proliferating phenotype[35], we used in vivo EdU labeling[36] to assess tumor cell proliferation. Notably, we found that Dll1+ cells were EdU negative, indicating reduced proliferation compared to Dll1− cells (Fig. 3f), further supporting the TIC-like characteristics of Dll1+ cells. Together, our studies identify Dll1+ cells as potential TICs with reduced proliferative potential in breast cancer.

**Dll1-mediated Notch signaling activates the NF-κB pathway in breast tumors**. To better understand the molecular signatures between Dll1+ and Dll1− tumor cells in breast cancer, we performed RNA-seq analysis using Dll1+ and Dll1− tumor cells from both PyMT-Dll1mCh and PyMT-Dll1GFP tumors (Supplementary Fig. 5a). As expected, signatures linked to stem cells, invasion and quiescent cells were enriched in Dll1+ tumor cells (Fig. 4a, b and Supplementary Fig. 5b, c), supporting our experimental data that Dll1+ cells have stem-like properties that increase their invasive properties. NF-κB and Hypoxia signaling are intimately linked to chemoresistance through regulation of cell proliferation and survival in several cancers, including breast cancer[37–40]. Next, we found activation of NF-κB, hypoxia and Notch pathways enriched in Dll1+ tumor cells (Fig. 4c and Supplementary Fig. 5d, e), which may contribute to the chemoresistance. Finally, we also found increase in metastasis signature to be enriched in Dll1+ tumor cells (Fig. 4d), supporting our in vivo metastasis data (Supplementary Fig. 4a–f). As expected, RNA-seq data demonstrated a ~7.5 higher level of

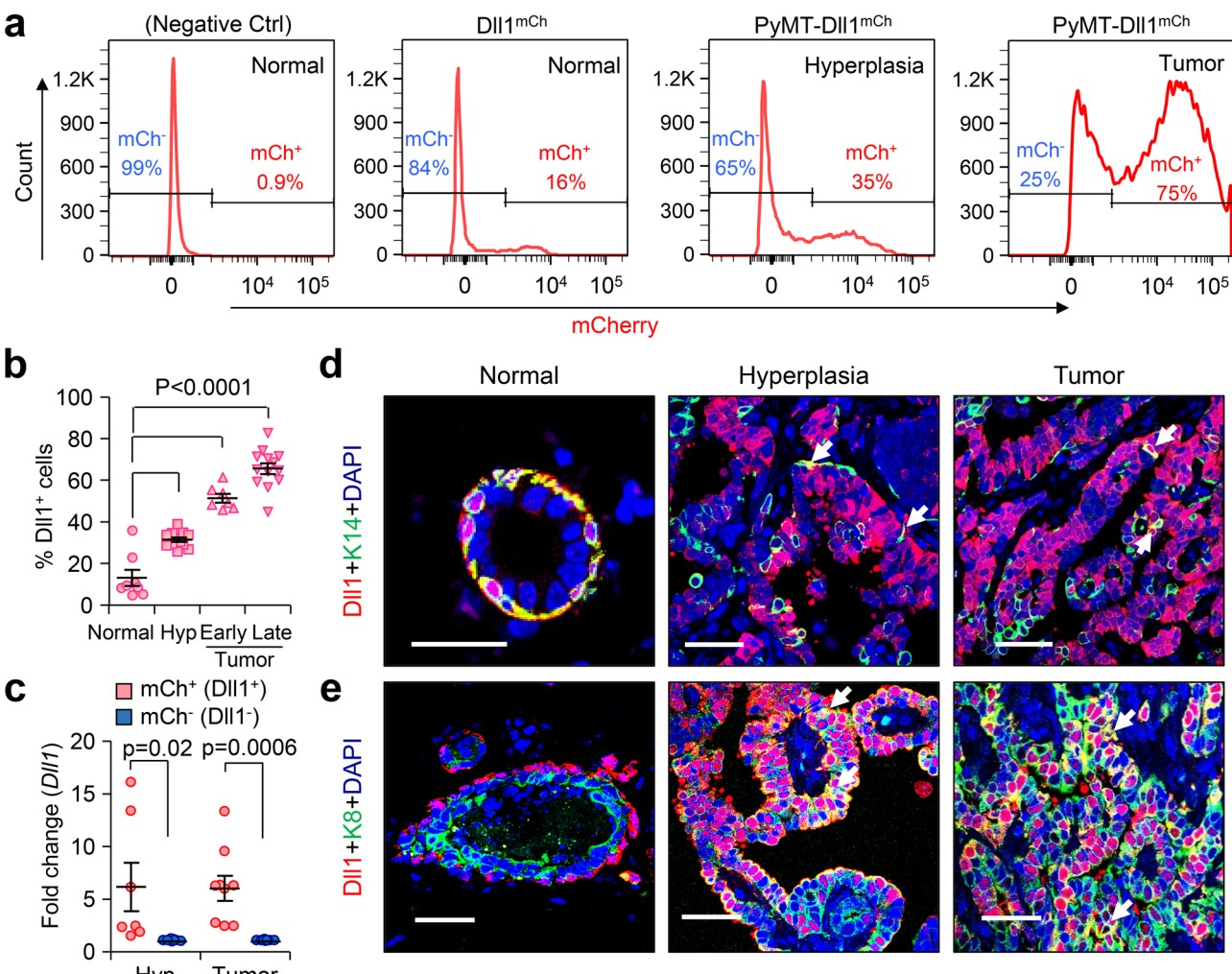

**Fig. 2 Number of Dll1+ luminal cells increases during PyMT tumor development. a, b** Flow cytometry profile of Lin⁻ tumor cells show increasing Dll1+ cells (mCherry expression) in PyMT-Dll1mCh tumors between normal to tumor stage, which are quantified in **b**. **b** $n = 8$ normal mammary gland, $n = 12$ hyperplatic mammary glands, $n = 7$ early tumors, and $n = 13$ late tumors. **c** qPCR analysis shows high *Dll1* mRNA expression in mCherry+ vs. mCherry⁻ tumor cells at hyperplasia and tumor stages ($n = 7$ hyperplastic mammary glands and $n = 9$ tumors). We enzymatically digested PyMT-Dll1mCh normal mammary glands, hyperplastic mammary glands, and tumors to sort lineage-negative (CD45⁻CD31⁻TER119⁻) Dll1+ or Dll1⁻ cells based on mCherry expression to quantify *Dll1* mRNA levels. Dll1⁻ (mCh⁻) fold considered 1 to calculate Dll1+ (mCh+) fold changes. **d–e** Representative immunofluorescence (IF) images of the normal mammary gland, hyperplasia, and tumor of PyMT-Dll1mCh mice show cellular distribution of Dll1 during tumor development. mCherry antibody was used to detect Dll1mCh+ cells. **d** White arrows indicate positive cells for a basal marker K14 and Dll1. **e** White arrows indicate the colocalization of Dll1 and K8 ($n = 3$ tumors per group). Please see more representative images in the supplementary figure 11. **b, c** P values were calculated using one-way ANOVA with Tukey's multiple-comparisons post-hoc test (**b**) and two-way ANOVA with Bonferroni post-test adjustment **c**. IF staining was done in three independent experiments using tumors from six independent experiments **d, e**. **b, c** Data are presented as the mean ± SEM. Scale bars, 40 μm (**d, e**). Source data are provided as a source data file.

DLL1 expression in this experimental setup. Considering a stringent cut off of >5.5 fold change gene expression in Dll1+ vs. Dll1⁻ tumor cells, we identified a set of 10 genes that are strongly expressed in Dll1+ cells as compared to Dll1⁻ cells (Supplementary Fig. 5f). Notably, 8 out of these 10 genes are related to lung and bone metastasis[41–48], suggesting that they could represent potential biomarkers.

To further investigate the mechanistic basis of Dll1-mediated effect on metastasis and cell survival of tumor cells, we performed ATAC-seq to examine the chromatin accessible and restricted regions in Dll1+ and Dll1⁻ tumor cells (Supplementary Fig. 5a). On a very global level, ATAC-seq data do not indicate any dramatic, epigenome-wide changes in chromatin accessibility patterns between Dll1+ and Dll1⁻ cells. This conclusion is based on finding similar numbers of UTR, exon, intergenic, intron, and

promoter peaks in both Dll1+ and Dll1⁻ tumor cells (Supplementary Fig. 5g). This finding is rather expected, given that these cells are of the same lineage, and as Dll1 is not itself a chromatin-modifying protein. We next performed an integrated analysis with ATAC-seq and RNA-seq data. For the integrated analysis, we took the "ATAC-seq Up" signature (all genes with a proximal ATAC-seq peak increasing significantly at p-adjusted < 0.05 in Dll1+ samples) and tested it for enrichment in the GFP/mCherry Up (Dll1+) combined RNA-seq ranked list. We found that this ATAC-seq signature is strongly enriched in the RNA-seq data (NES = 2.24, $p < 0.001$) (Fig. 4e). We then focused analyses on the 23 GSEA "Core Enrichment" genes (Fig. 4e) that are both in the ATAC-seq Up signature, and transcriptionally upregulated in Dll1+ cells using EnrichR online software. Our data shows that of these 23 common upregulated/increased accessibility genes from

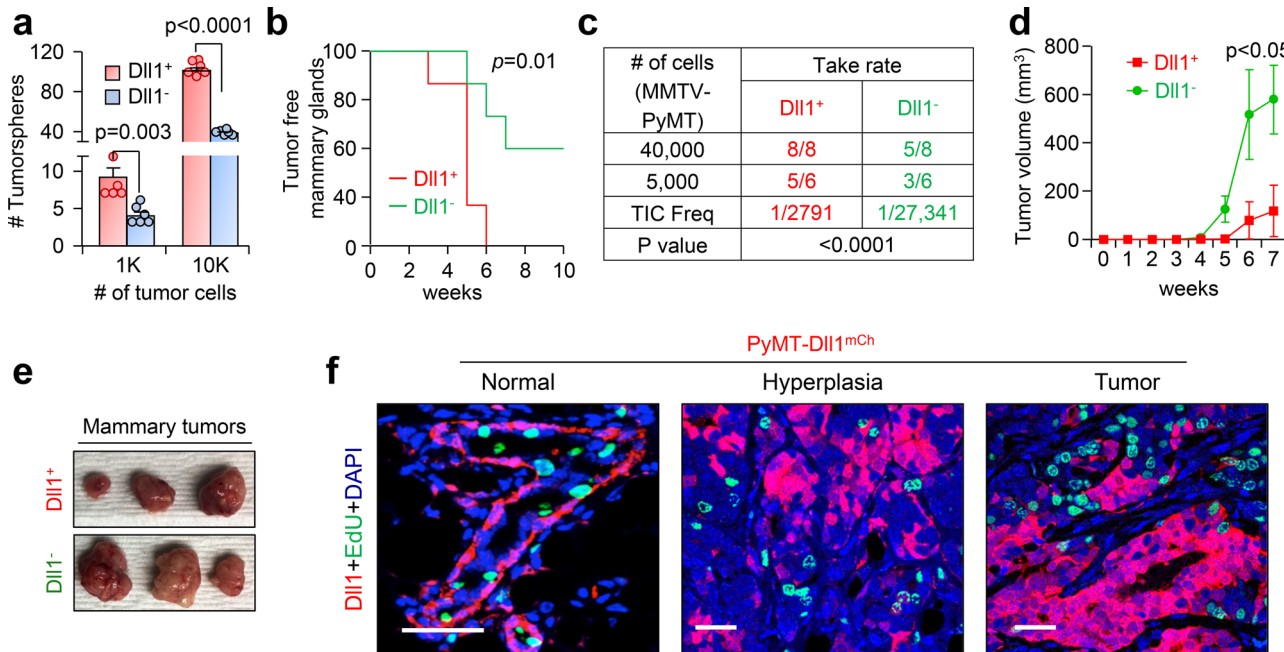

**Fig. 3 Dll1$^+$ cells are quiescent TICs and are highly metastatic compared to Dll1$^-$ tumor cells. a** A bar graph depicts the number of total tumorspheres from single sorted Dll1$^+$ and Dll1$^-$ tumor cells. Single cells were cultured in the low adherent plates for seven days in tumorsphere media[80] (n = 6 wells per group). **b–f** Dll1$^+$ (mCherry$^+$) and Dll1$^-$ (mCherry$^-$) tumor cells isolated from PyMT-Dll1$^{mCh}$ spontaneous tumors were injected into the mammary fat pad (MFP) of C57BL/6 mice. Contralateral mammary glands (4th position) were used for MFP injection, n is indicated in the table. **b** The KM-plot depicts the tumor initiation capacities of PyMT-Dll1$^+$ and PyMT-Dll1$^-$ tumor cells. The tumors were considered initiated when tumors grew bigger than 1×1 mm in dimension. **c** The table shows the tumor-forming potential of PyMT-Dll1$^+$ and PyMT-Dll1$^-$ tumor cells upon transplantation into C57BL/6 mice. **d, e** Representative growth curves and images from **c** show that Dll1$^-$ tumors grew faster than Dll1$^+$ tumor cells (n = 5 Dll1$^-$ and n = 8 Dll1$^+$ tumors). **f** Representative IF images illustrate Dll1$^+$ cells (mCherry) do not colocalize with EdU$^+$ proliferating cells (green) in the normal mammary gland, hyperplasia, and tumor of PyMT-Dll1$^{mCh}$ mice. mCherry antibody was used to detect Dll1$^{mCh+}$ cells. Two-way ANOVA with Bonferroni post-test adjustment **a, d**, two-tailed Log-Rank test **b** and Chi-square test **c** were used to calculate P values. **f** The IF was done twice using tumors (n = 3 samples/groups) from three independent experiments. **a, d** Data are presented as the mean ± SEM. Scale bars, 40 μm (**f**). Source data are provided as a source data file.

Dll1$^+$ cells, there are significant enrichment of both Notch signaling and NF-kB signaling (e.g., RelA) (Supplementary Fig. 5h), suggesting that Dll1-mediated Notch signaling may regulate NF- kB1 activity and signaling.

ATAC-seq data also identified the presence of enriched peaks of several of the prometastatic genes identified in the RNA-seq data from Dll1$^+$ cells (volcano plot, Fig. 4f). Dll1$^+$ tumor cells also have *NF-kB1* specific open chromatin region/enriched peak compared to Dll1$^-$ cells (Fig. 4g). Together, these data indicate that Dll1 activates NF-kB1 signaling in Dll1$^+$ tumor cells, potentially contributing to protumor survival and associated chemoresistance. This observation was further confirmed by an IHC analysis, where Dll1$^+$ tumors showed greater nuclear translocation of NF-kB compared with Dll1$^-$ tumors suggesting activation of NF-kB signaling (Fig. 4h, i).

To explore the clinical correlation of our mouse data, we examined our Dll1$^+$ mouse signature with METABRIC, a patient database to analyze correlation of *DLL1* to *NF-kB* in luminal A and B breast cancer patients[49,50]. Notably, we found that genes from DLL1$^{high}$ patient tumors show strong enrichment to Dll1$^+$ signature from PyMT$^{GFP}$ tumors or PyMT$^{mCh}$ (top 100 upregulated genes) (Fig. 4j and Supplementary Fig. 6a), highlighting the close resemblance of the mouse model to patients. Also, when patients were stratified based on *DLL1* expression, the *DLL1$^{high}$* tumors from patients had upregulated *NF-kB* signatures similar to mouse of RNA-seq data (Fig. 4k, l and Supplementary Fig. 6b, c). Moreover, tumors from luminal A and B patients show a strong positive correlation between *DLL1* and *NF-kB* (R = 0.185, p < 0.001, Fig. 4m). Finally, we found NF-kB motif genesets to be significantly enriched in *DLL1$^+$* patient tumors (Luminal A

and B) compared to *DLL1$^-$* patients (Supplementary Fig. 6d) using GSEA C3 compendium. Together, these observations suggest that Dll1-mediated Notch signaling induces activation of NF-kB signaling by regulating NF-kB1, which may contribute to tumor cell survival.

**Dll1$^+$ TICs are chemotherapy resistant**. TICs are thought to be responsible for resistance to chemotherapies in various cancers[51,52]. Therefore, we next assessed sensitivity to doxorubicin, an anthracycline chemotherapeutic drug effective in the treatment of a wide range of malignancies, including breast cancer[53,54]. To functionally test whether Dll1$^+$ cells are resistant to doxorubicin in the PyMT-Dll1$^{mCh}$ reporter model, we sorted out Dll1$^+$ and Dll1$^-$ tumor cells and treated them in vitro with doxorubicin. We found that Dll1$^+$ cells were resistant to doxorubicin-induced cell death compared to Dll1$^-$ cells (Supplementary Fig. 7a, b). Similar to our in vitro studies, treatment of transplanted Dll1$^+$ and Dll1$^-$ tumors with doxorubicin revealed that Dll1$^+$ tumors were resistant, whereas Dll1$^-$ tumor growth was significantly impaired upon doxorubicin treatment (Fig. 5a–d). Particularly, doxorubicin treatment significantly increased the number of quiescent/slow growing Dll1$^+$ TICs in tumors (Fig. 5e and Supplementary Fig. 7c), potentially due to their evasion of cell death and/or DNA damage. Indeed, further analysis of transplanted tumors revealed that Dll1$^+$ tumors were less prone to cell death compared to Dll1$^-$ tumors as seen by cleaved-caspase-3 and TUNEL staining (apoptotic cells) in the tumor cells (Fig. 5f–h). Cell cycle analysis further demonstrated a strong increase in a sub-G1 population (apoptotic cells) of Dll1$^-$

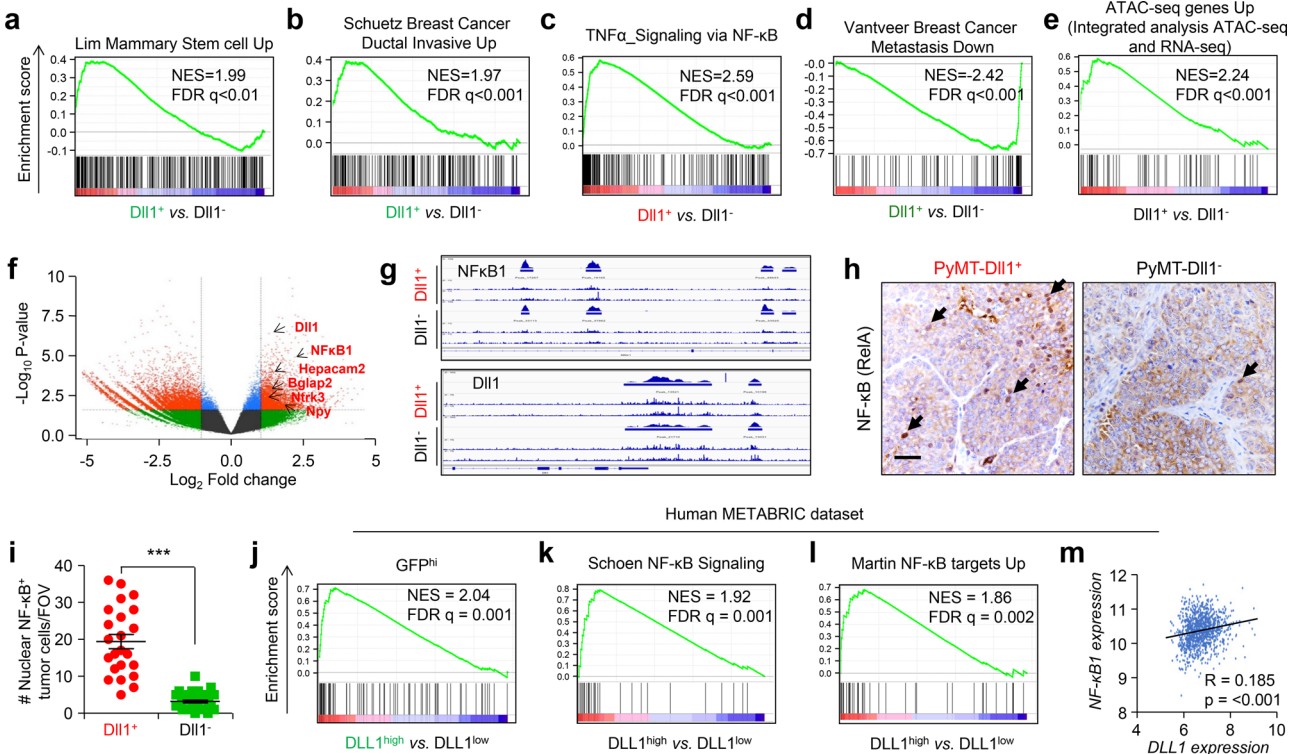

**Fig. 4 RNA-seq and ATAC-seq shows that Dll1+ tumor cells have enriched signatures for NF-κB pathway. a–e** We isolated Lin− Dll1+ and Dll1− tumor cells from either PyMT-Dll1GFP (Green color) or PyMT-Dll1mCh (Red color) tumors and performed a RNA-seq analysis. Geneset enrichment analyses (GSEA) demonstrate increased mammary stem cell **a**, increased invasiveness **b**, increased NF-κB signaling **c** and increased metastasis **d** signatures in Dll1+ tumor cells compared to Dll1− tumor cells. **e** Genes with a proximal ATAC-seq peak increasing significantly at $p < 0.05$ in Dll1+ cells show enrichment in the GFP/mCherry UP (Dll1+) combined RNA-seq ranked list. **f** Volcano plot from ATAC-seq analysis depicts enriched peaks for several protumorigenic and prometastatic genes in Dll1+ tumor cells compared to Dll1− cells. **g** NF-kB1 and Dll1 showed several open chromatin enriched peaks in Dll1+ tumor cells compared with Dll1− tumor cells. **h–i** Representative IHC images show higher nuclear NF-kB+ tumors cells in Dll1+ compared with Dll1− tumors. Cell sorting was performed to enrich for these population followed by mammary fat pad injection. The dots in scatter plot represents the FOV, which is obtained from $n = 3$ tumors per group. FOV is field of view. Black arrows indicate nuclear staining of NF-κB in tumor cells. **j** GSEA analysis of GFP+ tumor cells show enrichment of Dll1+ mouse tumor genes to genes of DLL1high luminal A and B patients (top 100 genes). **k, l** GSEA analyses show NF-κB signatures up in DLL1high luminal A and B patients compared with DLL1low luminal A and B patients. **m** Correlation plot depict a positive correlation between NF-κB1 and DLL1 expression in luminal A and B patient samples. **j–m** DLL1high and DLL1low patient data was obtained from METABRIC dataset ($n = 1140$ patient tumors). PyMT-Dll1mCh model was used for both RNA-seq and ATCT-seq whereas PyMT-Dll1GFP was used only for RNA-seq ($n = 2$ Dll1+ and $n = 2$ Dll1− Lin− tumor cells from PyMT-Dll1mCh tumors for RNA-seq were used; $n = 2$ Dll1+ and $n = 2$ Dll1− Lin− tumor cells from PyMT-Dll1GFP tumors for RNA-seq were used; and $n = 4$ Dll1− and $n = 3$ Dll1+ Lin− tumor cells isolated from PyMT-Dll1mCh tumors for ATAC-seq). Two-tailed unpaired Student's t-test (**i**) and pearson correlation coefficient **m** were used to calculate P values. **i** Data are presented as the mean ± SEM. Scale bar, 60 μm **h**. Source data are provided as a source data file.

cells compared to Dll1+ tumor cells (Supplementary Fig. 8a, b). Finally, γH2AX staining (green staining indicates DNA damage) shows reduced doxorubicin-induced DNA damage in Dll1+ tumor cells compared to Dll1− tumors (Supplementary Fig. 8c–e), potentially explaining the chemotherapeutic resistance of Dll1+ tumors (Fig. 5b–d).

**Pharmacological inhibition of Dll1 sensitizes Dll1+ tumor cells to chemotherapy.** Next, we investigated whether targeting Dll1 in vivo could impact the chemoresistant phenotype of Dll1+ TIC/CSCs. Dll1+ tumor-bearing mice were given either the control vehicle or anti-Dll1-blocking antibody (αDll1)[55] or doxorubicin (Dox), or a combination of both (Supplementary Fig. 9a). In this tumor initiation and progression study, Dll1 antibody alone had a modest inhibitory effect on tumor growth, whereas, the combination treatment dramatically reduced tumor growth (Fig. 6a, b), suggesting that Dll1-blocking sensitizes Dll1+ TICs to chemotherapy. Moreover, metastatic events were significantly decreased in mice treated with anti-Dll1 antibody and

chemotherapy compared with untreated controls (Fig. 6c–e and Supplementary Fig. 9b). This decrease in tumor growth and progression was associated with reduced nuclear translocation of NF-kB (Fig. 6f, g), suggesting that Dll1-mediated chemoresistance is potentially driven by NF-kB signaling.

There were no observable side effects of Dll1 blocking in mono or in combination with doxorubicin, as body weight of these mice were unaltered (Supplementary Fig. 10a). The activity of the Dll1 antibody was further confirmed by examination of depletion of marginal B-cells in the spleen that depend on Dll1 activity (Supplementary Fig. 10b, c)[56–58]. Taken together, these results delineate that Dll1 targeting may be effective for patients failing chemotherapy and may have a better safety profile during treatment.

**Pharmacological inhibition of NF-kB and Dll1 resensitizes chemoresistant Dll1+ luminal tumor cells, phenocopying the effects of Dll1-blocking antibody.** Since NF-kB was found to be downstream of Dll1-mediated Notch signaling, we next sought to

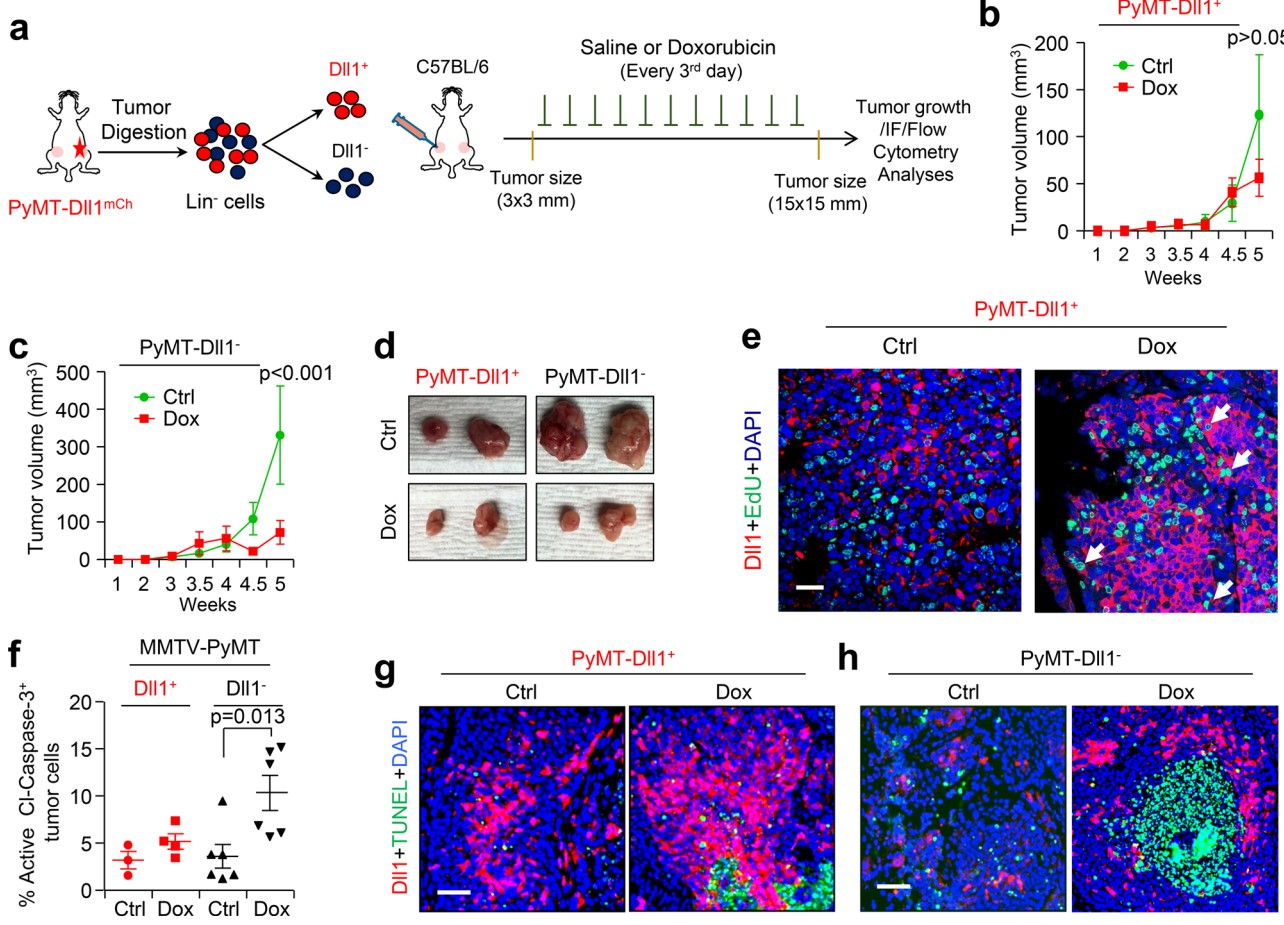

**Fig. 5 Dll1$^+$ tumor cells are chemoresistant in PyMT tumors. a** Schematic shows the experimental plan of in vivo doxorubicin sensitivity assays using PyMT-Dll1$^+$ and PyMT-Dll1$^-$ tumors. **b, c** Tumor growth curves show PyMT-Dll1$^+$ **b** and PyMT-Dll1$^-$ **c** tumor growths upon saline (ctrl) or doxorubicin treatments. **d** Representative whole tumor images from **b**, **c**. **b–d** $n = 6$ tumors/group. **e** The representative IF images show the expression of Dll1 (Red) and EdU$^+$ proliferating cells (Green) in PyMT-Dll1$^+$ tumors upon treatment of saline or doxorubicin. mCherry antibody was used to detect Dll1$^{mCh+}$ cells. The white arrows indicate proliferating Dll1$^+$ EdU$^+$ tumor cells upon doxorubicin treatment. **f** The flow cytometry data demonstrate the percentage of active cleaved-caspase-3 positive cells in PyMT-Dll1$^+$ and PyMT-Dll1$^-$ tumors treated with either saline (ctrl) or doxorubicin. **g, h** The representative IF images show the expression of Dll1 (Red) and TUNEL$^+$ apoptotic cells (Green) in PyMT-Dll1$^+$ **g** and PyMT-Dll1$^-$ **h** tumors upon treatment of saline or doxorubicin. The formalin-fixed tumors specimen were stained with the mCherry antibody to detect Dll1$^+$ tumor cells. $P$ values were calculated using two-way ANOVA with Bonferroni post-test adjustment **b**, **c** and one-way ANOVA with Tukey's multiple-comparisons post-hoc test **f**. IF experiment was repeated twice using $n = 3$ tumors/group **e** and once with $n = 4$ tumors/group **g**, **h**. Data are presented as the mean ± SEM. Scale bars, 60 μm **e**, **g**, **h**. Source data are provided as a source data file.

determine if the NF-kB inhibitor IMD-0354 (IMD) would resensitize tumor cells to chemotherapy, phenocopying the effect of the Dll1-blocking antibody. IMD is a selective IKK2 inhibitor, which disrupts NF-κB signaling, which showes activities in several preclinical models including cancer[59–62]. An in vitro tumorsphere assay with Dll1$^+$ tumor cells demonstrated a significant decrease in the number and size of tumorspheres with a combination treatment of IMD and doxorubicin, similar to effects seen with Dll1-blocking antibody and doxorubicin (Fig. 7a–c). Especially, combination treatment of Dll1 antibody, IMD and chemotherapy further decreased the number and size of tumorspheres (Fig. 7a–c), while either IMD or chemotherapy alone had minimum effect on tumor cells. To complement the in vitro data, we treated Dll1$^+$ tumor cells with doxorubicin, Dll1-blocking antibody, IMD-0354 treatment alone or in combinations (Supplementary Fig. 10d). Similar to in vitro results, NF-kB inhibition along with αDll1 ab or doxorubicin was very effective to inhibit tumor growth with no significant toxicity (Fig. 7d, e and Supplementary Fig. 10e). Furthermore, combination treatment

targeting Dll1, NF-kB and doxorubicin significantly abolished tumor growth of Dll1$^+$ tumor cells. Thus, our results suggest that activation of NF-kB downstream of Dll1 signaling in tumor cells promotes chemoresistance and blocking all these components together can lead to significant response in thes tumors.

## Discussion

Notch signaling, specifically the Notch receptors, has been associated with chemoresistance in several cancers, including breast cancer, most markedly in TNBC[63–68], suggesting that targeting this pathway may improve clinical outcome. Our study reveals a central role for the Notch ligand Dll1 in promoting tumor progression, metastasis and chemoresistance in aggressive luminal breast tumors. Identification of a ligand-based therapy to block tumor-specific Notch signaling may provide a safer and effective therapeutic option for breast cancer patients.

Earlier studies from our group have demonstrated that the Notch ligand Dll1 is important for normal mammary stem cell

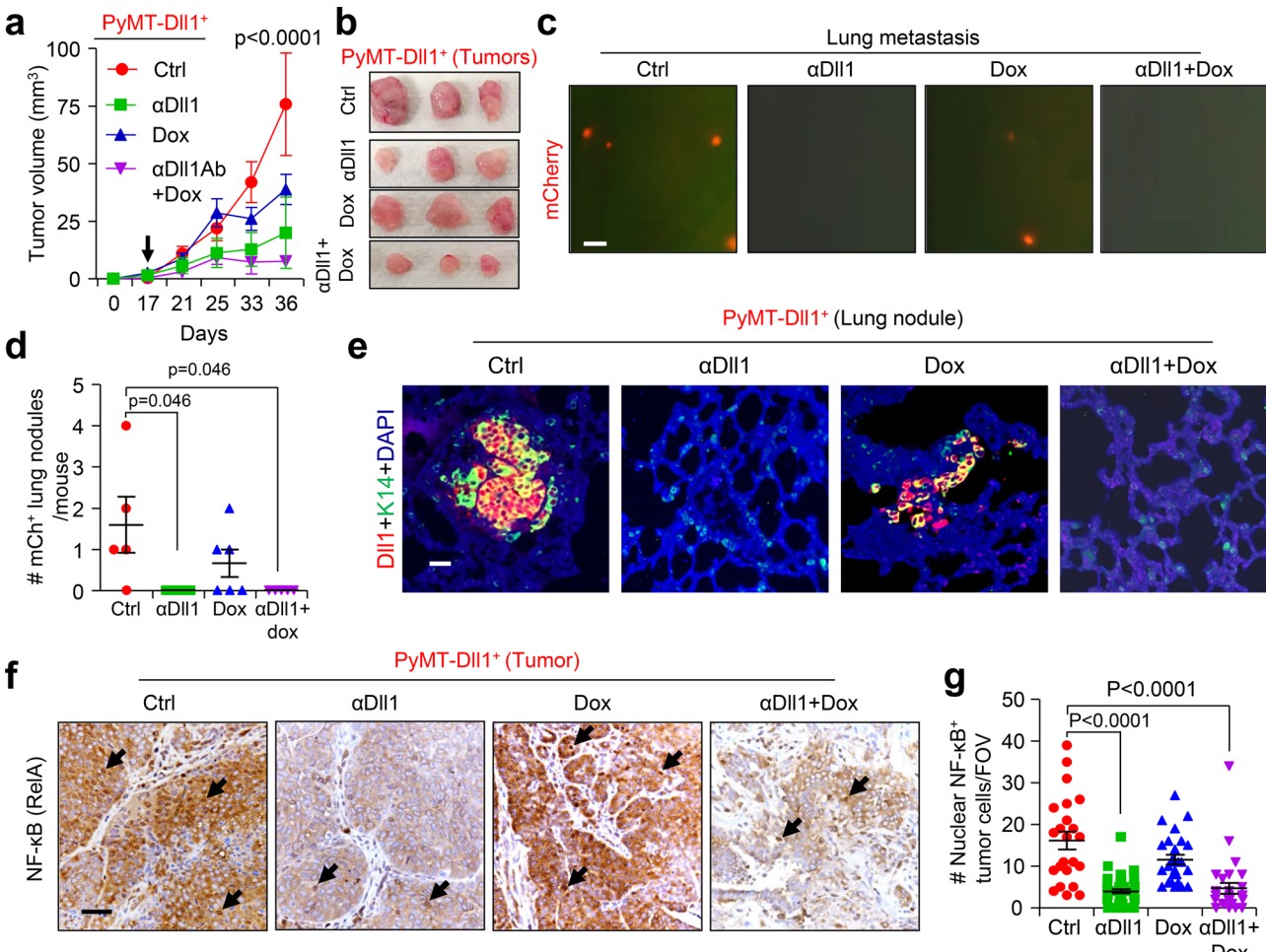

**Fig. 6 The Dll1-blocking antibody treatment sensitizes the Dll1+ tumor cells to the chemotherapy. a, b** Tumor growth curves **a** and representative whole tumor images (**b**) after Dll1+ tumor cells derived tumors were treated with indicated treatments (more details are in Supplementary Fig. 9a) ($n = 4$ for ctrl, αDll1 ab, and αDll1 ab +Dox, $n = 6$ Dox tumors). The black arrow indicates the day when the treatments started. **c–e** Representative fluorescence images show mCherry+ nodules in whole lungs with quantification **c, d**. The scatter plot in **d** shows mCherry positive lung nodules, which were quantified under fluorescent dissection microscope after harvest (data combined from two independent experiments, $n = 5$ Ctrl, $n = 5$ αDll1 ab, $n = 6$ Dox, and $n = 5$ αDll1 ab + Dox tumors). **e** Representative IF images of lung cross sections show presence of mCherry+ cells (red). mCherry antibody was used to detect Dll1mCh+ cells. **f, g** The representative IHC images show the NF-κB staining in Dll1+ tumors upon indicated in vivo treatments, which is quantified in **g**. The dots in scatter plot represents the FOVs, which is obtained from $n = 3$ tumors per group. FOV is field of view. Black arrows indicate nuclear staining of NF-κB in tumor cells. $P$ values were calculated using two-way ANOVA with Bonferroni post-test adjustment **a** or one-way ANOVA with Tukey's post-hoc test **d, g**. Data present three **c, e** or two **f** independent experiments. Data are presented as the mean ± SEM. Scale bars, 200 μm **c**, 20 μm **e** and 60 μm **f**. Source data are provided as a source data file.

development[25]. We further showed that Dll1 is preferentially upregulated in luminal breast cancer compared to TNBC/basal cancer and regulates CSC function in these breast cancer cells[24]. However, at that time, we could not evaluate the spatial and temporal distribution of Dll1 expression during tumor progression and metastasis. Furthermore, we could not determine if Dll1+ cells themselves had TIC/CSC properties due to the lack of appropriate mouse models. In this study, we used Dll1 conditional knockout and reporter models crossed with spontaneous murine models (MMTV-PyMT and MMTV-Wnt1) to determine the spatial and temporal distribution of Dll1 and also determine the function of Dll1+ tumor cells in tumor progression. We found that Dll1+ cells increase dramatically from normal mammary gland to hyperplasia and tumor, suggesting a tumor supporting function of Dll1. We found that Dll1+ tumor cells increase in K8+ luminal compartment compared to basal cells as tumor progresses from hyperplasia to tumor stage in MMTV-PyMT model. Future studies on lineage

tracing starting at earlier age in mice, will help to decipher the mechanism behind this increase in K8+ Dll1+ luminal cells in tumors. Using conditional-knockout mice of Dll1, we further demonstrated that Dll1 has a significant tumor-promoting function in MMTV-PyMT tumors but not in MMTV-Wnt1 tumors. It is possible that Dll1 may have tumor-promoting function in other basal tumors, which needs further explorations. We further show that Dll1+ tumor cells are quiescent TICs and Dll1-mediated Notch signaling is one of the major drivers of tumor progression and metastasis in the the MMTV-PyMT model using both reporter and conditional-knockout mouse models. These data are in accordance with the Metastatic Breast Cancer project database (Broad Institute), which highlights the amplification of Dll1 in metastatic breast cancer patients.

Tumor heterogeneity is currently a clinical challenge for breast cancer therapy[11,69–71]. To determine if the Dll1+ tumor cells resemble TICs that could increase tumor burden, metastasis, and

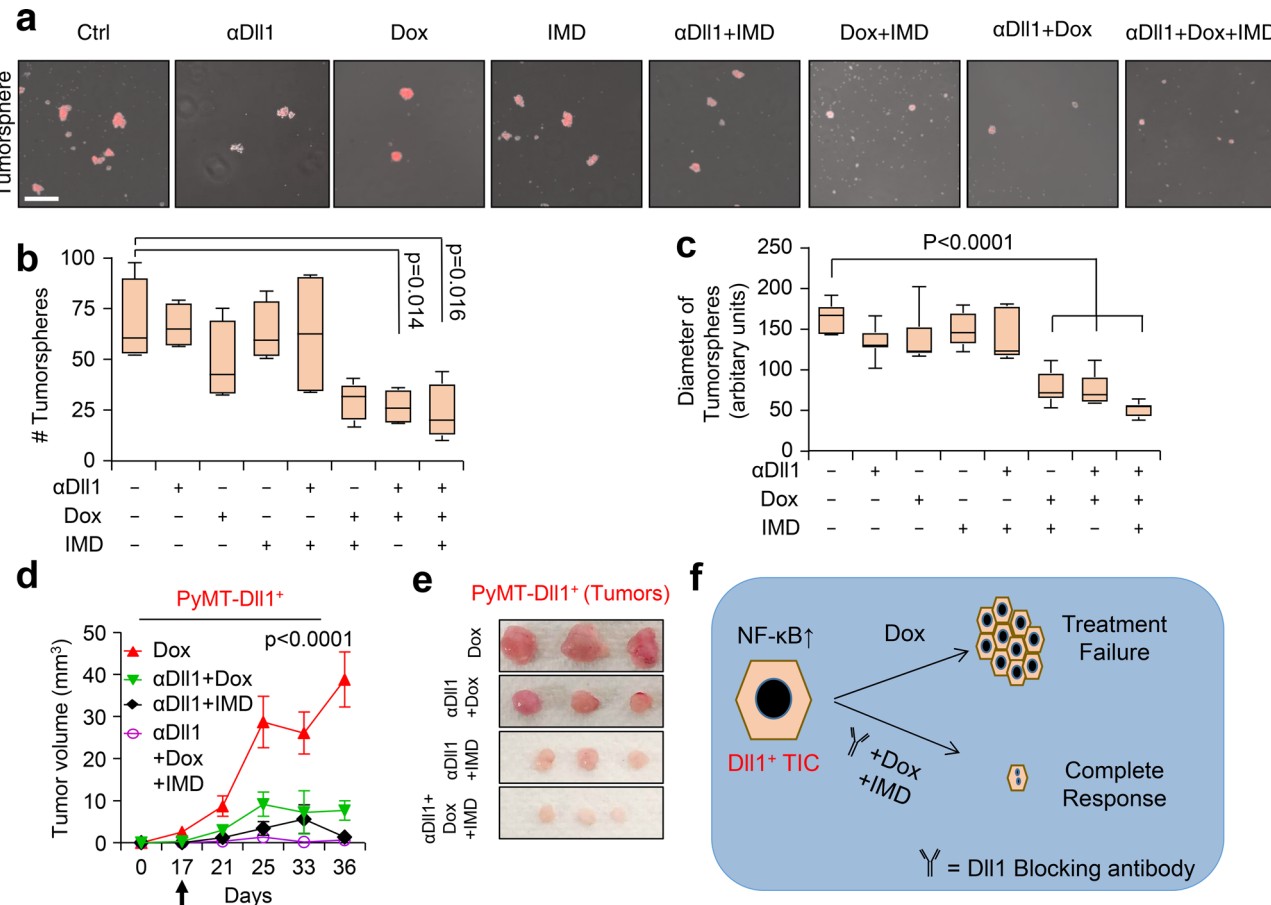

**Fig. 7 Inhibition of NF-κB sensitizes Dll1+ tumor cells to chemotherapy. a** Representative fluorescence images show the tumorspheres of Dll1+ tumor cells with the indicated treatments. (IMD 100 nM, doxorubicin 80 nM, and Dll1 antibody 500 μg/ml). For combination treatment, 50 nM of IMD and 40 nM of doxorubicin were used. **b, c** The box plots show the quantification of tumorspheres in number **b** and diameter (**c**). The boxes represent the 75th, 50th, and 25th percentile of the values. The top and bottom lines represent maximum and minimal data points within the ×1.5 IQ (interquarter) range, respectively. **d** Tumor growth curves of tumors derived from Dll1+ tumor cells with indicated treatment (more details are in Supplementary Fig. 10d). The black arrow indicates the day when the treatments started. Representative whole tumor images are shown in the **e**. The alone doxorubicin and αDll1+Dox groups were used for both the experiments in Fig. 6a, b and 7d, e (n = 6 Dox, n = 4 αDll1 ab+Dox, n = 4 αDll1 ab+IMD, n = 6 αDll1 ab+Dox+IMD tumors). **f**, The model shows that PyMT-Dll1+ TICs are resistant to the doxorubicin treatment, along with increased nuclear translocation of NF-κB, which can be sensitized by blocking Dll1 using antibody and NF-κB inhibitor (IMD) along with doxorubicin. The combination treatment achieves a almost complete response in preclinical mouse models. *P* values were calculated using one-way ANOVA with Tukey's post-hoc test (**b, c**) and two-way ANOVA with Bonferroni post-test adjustment (**d**). Data present two (**a–c**) independent experiments. Data are presented as the mean ± SEM. Scale bars, 100 μm (**a**). Source data are provided as a source data file.

chemotherapeutic resistance, the transcriptional signatures of Dll1+ and Dll1− tumor cells were further evaluated by RNA-seq analysis. These data show that Dll1+ tumor cells are associated with increased invasiveness and metastatic signatures in GSEA, corroborating experimental data. The RNA-seq data indicated that Notch target genes are activated in Dll1+ tumor cells compared with Dll1− tumor cells. However, we need further evaluations to understand Notch signalling between Dll1+ cells and Notch receptor-expressing cells. Moreover, Dll1+ tumor cells are enriched for stem cell genes and as expected, the Dll1+ cells showed reduced expression of cell cycling genes, while Dll1− cells were highly enriched, highlighting the relative quiescent properties of Dll1+ tumor cells, which is confirmed by EdU assay. Previous studies suggest that CD44hi/CD24lo epithelial cells retain classic TIC features in breast cancer, including tumor-initiating capacity in vivo, tumorsphere formation, and resistance to standard chemotherapy[13]. We found that Dll1+ tumor cells are resistant to the chemotherapeutic drug, doxorubicin. Mechanistically, we showed that Dll1+ tumor cells resist chemotherapy

by avoiding drug-induced cell death and DNA damage as seen by cleaved-caspase-3 assay, γH2AX staining, TUNEL and cell cycle assays. Interestingly, Dll1+ cells were also enriched for signatures of NF-κB signaling and hypoxia, common features of cancer progression and chemoresistance[72–75]. This observation was further confirmed in human patients data using METABRIC dataset, where we found strong positive correlation between *DLL1* and *NF-κB1*. Moreover, Dll1high gene signatures (top 100 genes) from our mouse models closely resemble genes of DLL1high Luminal A and B patients supporting the clinical significance of our model to human patients. Indeed, the NF-κB pathway can crosstalk with Notch signaling in lung cancer to promote chemoresistance[76]. Moreover, Hypoxia has long been associated with chemoresistance to several drugs, including doxorubicin, in multiple cancers such as lung, prostate, and breast cancer[39,77–79]. ATAC-seq corroborates RNA-seq data and identified several open chromatin regions for the *NF-κB1* gene and other protumorigenic and prometastatic genes, indicating that Dll1-mediated Notch signaling may regulate these pathways to resist chemotherapy.

Furthermore, increased nuclear translocation of NF-κB was observed in tumor cells expressing higher levels of Dll1. Blocking of NF-kB activity using the IMD inhibitor in combination with doxorubicin significantly decreased tumor cells viability and stem cells activity in tumorsphere assays, thus phenocopying the effects of the Dll1-blocking antibody. Combination treatment blocking NF-kB activation, Dll1 and chemotherapy shows almost complete respone in tumor growth in vivo.

Together, our data suggest that Dll1-mediated Notch signaling drives metastasis and chemoresistance and may therefore be responsible for higher mortality in breast cancer (Fig. 7f). Mechanistically, our data suggest that Dll1 increases resistance to DNA damage and cell death through the NF-κB pathway. Remarkably, combination therapy targeting Dll1 ligand and NF-κB pathway shows complete response to the doxorubicin in Dll1$^+$ tumor cells. Altogether, our findings demonstrate a promising therapeutic target in breast cancer patients with advanced disease and metastasis, particularly for those who are chemoresistant. Further preclinical studies will assess the potential efficacy of this strategy in the clinics.

## Methods

**Animal studies**. Animal housing, handling, and procedures were conducted in compliance with the Institutional Animal Care and Use Committee (IACUC) of the University of Pennsylvania. The health status of all mice was normal, and they were not involved in any previous procedure. For Dll1 in vivo loss-in-function studies, Dll1$^{cKO}$ mice[25] were crossed with MMTV-PyMT or MMTV-Wnt1 to obtain PyMT-Dll1$^{cKO}$ or Wnt1-Dll1$^{cKO}$ mice. To obtain PyMT-Dll1$^{mCh}$ and PyMT-Dll1$^{GFP}$ mice, the Dll1$^{mCherry}$ BAC-based mice[30] and Dll1$^{GFP}$ knockin[25] mice were crossed with MMTV-PyMT mice. All mice were in C57BL/6 background for tumor studies. Six-week-old female wild-type C57BL/6 mice for mammary fat pad injections were obtained from Jackson Laboratory. For mammary fat pad injections, all mice were anesthetized, and tumor cells were injected into the mammary fat pad following established protocols[71]. Tumors were detected by palpation and measured twice a week. The experiment was terminated when tumors reached certain size as mentioned in the figure legend, or the experiment was terminated before if the tumors were ulcerated (humane endpoint). Mice were injected with 0.2 mg/10 g body weights EdU (Invitrogen, A10044) intraperitoneally 12 h before sacrifice to study in vivo proliferation of tumor cells.

**Metastatic lung nodule count**. The tumor-bearing mice were sacrificed by cervical dislocation to harvest the metastatic lungs. For PyMT-Dll1$^{cKO}$ and PyMT-Dll1$^{WT}$ mice, the lungs were fixed with Bouin's solution for 24 h and then washed with 70% ethanol and then counted for nodules based on whole lung images. For PyMT-Dll1$^{mCherry}$ and PyMT-Dll1$^{GFP}$ mice, after a brief wash with PBS, the mCherry$^+$ lung nodules were counted under Leica fluorescent dissection microscope. Lungs were also sectioned and the lungs sections were stained for H&E as per described protocol[24,80]. The H&E slides were evaluated by a board certified pathologist to confirm metastatic nodules.

**In vivo cell proliferation assay**. We intraperitoneally injected EdU (0.2 mg/10 g body weight) 12 h before harvesting the tumors. EdU-labeled proliferating cells were detected on Formalin-fixed tissues by Click-iT EdU Alexa Fluor 488 imaging kit (Thermo Fisher Scientific) following the manufacturer's protocol.

**Histological analysis and immunofluorescence (IF)**. For histological analysis, mouse tumor and mouse lung specimens were processed as previously described[81,82]. Antibodies information and dilutions are listed in Supplementary Table 2. DAPI was used to stain nuclei. IF Images were taken using a Leica SP5 FLIM Confocal microscope.

**Flow cytometry analysis and sorting**. The single-cell suspension for flow cytometry analysis of normal mammary glands, hyperplasia and tumors was obtained following a published protocol[71,82]. Single cells were stained with a combination of antibodies (listed in Supplementary Table 1) for 30 min on room temperature in the dark. Live cells were gated out using either DAPI or PI. Flow cytometry analysis was performed using the LSRII and LSRFortessa Flow Cytometers (BD Biosciences), and data were analyzed using FlowJo software (TreeStar, Inc). We used the Aria II instrument for sorting mammary tumors.

**Protein extraction and western blot analysis**. We digested MMTV-PyMT tumors into single cells to isolate protein. We performed western blot analysis as

described in previous articles[24]. The antibodies details and dilutions are listed in Supplementary Table 3.

**RNA extraction and quantitative PCR (qRT-PCR)**. Total RNA was isolated from sorted tumor cells or hyperplasia using the Invitrogen RNA extraction kit in accordance with the manufacturer's instructions. The amount of isolated RNA was quantified using a Nanodrop spectrophotometer (Thermo Scientific). Complementary DNA was synthesized from 1 μg of total RNA using SuperScript (Invitrogen). Real-time RT-PCR was performed on the Applied Biosystems StepOne Plus PCR machine (Thermo Fisher) using SYBR Green Power (Life Science Technologies). We performed all qRT-PCR assays in duplicate at least three independent experiments. The gene-specific primer sets were used at a final concentration of 0.2 μM and their sequences are listed in Supplementary Table 4.

**ATAC-seq library preparation and analysis**. Single-cell digestion from freshly isolated spontaneous mammary tumors from PyMT-Dll1$^{mCh}$ tumor-bearing mice was performed followed by cell sorting. Lineage-negative epithelial tumor cells, which are Dll1$^+$ and Dll1$^-$ cells (based on mCherry expression) were sorted. 20,000 Dll1$^+$ and Dll1$^-$ lineage-negative tumor cells from four different tumors were used. Previously described established protocols for ATAC-Seq transposition assay and library preparation were used[83,84].

Raw sequence (fastq) files were processed in genotype-specific replicate batches to find open chromatin regions using the ENCODE ATAC-seq pipeline https://github.com/ENCODE-DCC/atac-seq-pipeline. Peaks enumerated in the optimized peak files for the Dll1$^+$ and Dll1$^-$ replicate groups were merged with bedops[41], and the nucleotide sequence underlying each of the resulting regions was extracted to a fasta file using bedtools (https://bedtools.readthedocs.io/en/latest/). Reads for each sample were mapped to the sequences for all regions using salmon[85], and the read matrix was normalized and analyzed for differential peak depth between the two genotypes using DESeq2[86]. Peak locations were annotated for nearby genes using homer (http://homer.ucsd.edu/homer/ [87]) and merged into the RNA-seq dataset based on gene symbol. Data were visualized using the Integrative Genomics Viewer[88].

**RNA-sequence analysis and geneset enrichment analysis (GSEA)**. Raw sequence files (fastq) for eight samples were mapped using salmon (https://combine-lab.github.io/salmon/) against the mouse transcripts described in genecode (version M20, built on the mouse genome GRCm38.p6, https://www.gencodegenes.org), with an average of 51.3 M total input reads for each sample, and an 80.1% average mapping rate. Transcript counts were summarized to the gene level using tximport (https://bioconductor.org/packages/release/bioc/html/tximport.html), and normalized and tested for differential expression using DESeq2 (https://bioconductor.org/packages/release/bioc/html/DESeq2.html). A 2-factor statistical model was used including Dll1 status (+/−) and mouseID. Gene Set Enrichment Analysis (GSEA, http://software.broadinstitute.org/gsea/index.jsp) was run for the contrast in pre-ranked mode using the DESeq2 statistic of the ranking metric. Mouse gene symbols were mapped to human gene orthologs using Ensembl's BioMart (http://www.ensembl.org/biomart/martview/). Where there were redundant mappings, the statistic with the highest absolute value was chosen. Geneset enrichment analysis (GSEA) was conducted using GSEA2.2.4 software to generate enrichment scores for genesets in Hallmark, C2.all and C5.all, datasets with default settings.

**METABRIC dataset analysis**. RNA expression data from the METABRIC dataset[49,50] were downloaded from the cBio portal at http://download.cbioportal.org/brca_metabric.tar.gz, and corresponding patient clinical data were downloaded from https://www.cbioportal.org/study/clinicalData?id=brca_metabric. Luminal A and Luminal B patient subtypes were identified using the Pam50 + Claudin-low subtype parameter included in the clinical data. Pearson correlation was used to determine statistical significance of correlation of DLL1 expression and expression of NF-kB1 in the Luminal A and Luminal B combined patient cohort. The combined cohort was then stratified by median DLL1 expression, and GSEA was performed by testing all genesets within the GSEA C2 and C3 MSigDB compendia for enrichment in the list of genes ranked by magnitude of over-expression in DLL1$^{high}$ (above median) vs. DLL1$^{low}$ (below median) patients. The signal-to-noise (S2N) metric was used for gene ranking, and permutations ($n = 1000$) were performed on genesets. Default settings were used for all other GSEA parameters. To derive Dll1$^{high}$ and Dll1$^{low}$ experimental gene expression signatures from sorted mouse tumor cells, genes were ranked by magnitude of overexpression in RNA-seq data from Dll1$^{high}$ (GFP$^+$ or mCherry$^+$) vs. Dll1$^{low}$ (GFP$^-$ or mCherry$^-$) samples. The S2N metric was used for ranking genes, and the top 100 most overexpressed genes were used for Dll1$^{high}$ genesets.

**In vivo Dll1-blocking antibody, doxorubicin and IMD treatments**. The mice were randomly assigned to the cohorts for drug treatments. Treatments were started when tumor size reached 3 × 3 mm. Supplementary Figs. 9a and 10d provide complete experimental strategies. The Dll1-blocking antibody (18 mg/kg thrice a week), doxorubicin (1 mg/kg twice a week), and IMD (0.5 mg/kg once a week) were given until the endpoint of the experiment.

**MTT assay**. Dll1$^+$ and Dll1$^-$ tumor cells were obtained form PyMT-Dll1$^{mCherry}$ spontaneous tumors. The 10,000 cells were seeded in each well of 96-well plate. Forty-eight hours after seeding, the cells were treated with the 100 ng/mL of Doxorubicin for 4-days. After incubation, we added 20 uL of the MTT reagent (5 mg/mL) with 80 μL of MEGM media to each well. We incubated the plate in cell-culture incubator for 4 h. We added 100 μL of DMSO to solubilize the purple formazan crystals and the absorbance was observed at 600 nm wavelength and reference wavelength at 750 nm as described[89].

**Tumorsphere assay**. We digested the PyMT-Dll1$^{mCh}$ tumors into single cells and sorted Lin$^-$ Dll1$^+$ and Dll1$^-$ tumor cells. The tumor cells were cultured for tumorsphere formation, as previously described[81]. Briefly, cells were grown in Corning low adherent plates in tumorsphere media for 5–7 days[81]. The tumorsphere media includes 50 ng/mL EGF, 150 ng/mL FGF, 1× B27 in DMEM:F12 (1:1) media. The number of total tumorspheres and diameter were counted/measured using a microscope. The single cells from tumors were made following a published protocol[80].

#### Quantification and statistical analysis

*Sample size and replicates*. We repeated our in vitro experiments at least three times in triplicate. The exact number of mice for each group is mentioned in the corresponding figure legends. For in vivo treatments of Dll1-blocking antibody, doxorubicin and IMD, mice were randomized based on their age and body weight, before the start of the experiment.

*Replicates statistical analysis*. The results are reported as mean ± SEM (standard error of the mean). The significance of differences was calculated using two-tailed unpaired Student's $t$-test for normally distributed datasets. One-way ANOVA with Tukey's multiple-comparisons post-hoc test was used for multiple pairwise comparisons. The tumor growth and grouped datasets were analyzed using the Bonferroni corrected two-way ANOVA to compute statistical significance. Differences in tumor initiation between groups via Kaplan–Meier plots were statistically evaluated using Log-rank tests. For limiting dilution assay, Chi-square test and ELDA online software was used. The number of animals and the statistical tests to compute $p$-value are reported in each of the corresponding figure legends.

**Reporting summary**. Further information on research design is available in the Nature Research Reporting Summary linked to this article.

### Data availability

Sequencing data from RNA-sequencing and ATAC sequencing from tumor samples that support the findings of this study (Fig. 4 and Supplementary Figs. 5 and 6) are deposited in the Gene Expression Omnibus, reference GSE129198 (RNA-seq) and GSE133736 (ATAC-seq). Source data are provided with this paper.

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

## Acknowledgements

We thank Dr. Leslie King (University of Pennsylvania, PA) for careful reading of the manuscript and helpful discussions. We thank Dr. Michael L. Atchison and Dr. Sarmistha Banerjee for generously providing flow cytometry antibodies of marginal B-cells. We thank the animal facility located in Hill Pavilion building for housing and maintenance of all mouse models. We thank the Center for Host-Microbial Interactions for running the ATAC-Sequencing libraries. We thank Penn Vet Comparative Pathology Core for embedding and sectioning of tumor and lung tissues. We thank Penn Vet imaging core for providing confocal microscopy service. We thank Flow Cytometry core at Children's Hospital of Philadelphia and the University of Pennsylvania for FACS analysis and cell sorting. This work was supported by grants from Abramson Cancer Center Emerson Collective Fund, American Cancer Society, McCabe Fund Fellow Award from Perelman School of Medicine, UPENN and NCI-K22 grant to R.C. (K22CA193661) and American Cancer Society grant to R.C. (RSG DDC - 133604).

## Author contributions

S.K. and R.C. conceptualized the study and designed all experiments. S.K. and A.N. performed all the experiments. R.R. helped with genotyping of mice. M.A.B., S.S., and J.W.T. analyzed the RNA-sequencing data. J.W.T. and M.A.B. also analyzed the ATAC-sequencing data. A.K-S. performed histopathological evaluation of lung metastasis in the manuscript. M.A.B. analysed METABRIC dataset. C.L. and N.L. provided the kit and the support for ATAC-sequencing library preparation. C.W.S. provided Dll1-blocking antibody. A.L.W. provided helpful discussions. Y.K. provided Dll1 reporters and knockout mice. S.K. and R.C. wrote the manuscript. All authors discussed the results and commented on the manuscript.

## Competing interests

The authors declare no competing interests.
