## [Peer Review File · Nature Communications]

Reviewers' Comments:

Reviewer #1:

Remarks to the Author:

This work is based on a previous study from the same group (Chakrabarti et al, Science 2018), in which the authors used the Dll1-Cre mouse line as well as a Dll1-mCherry reporter mouse to characterise the Dll1+ cells in the normal mammary gland. Now the authors use the same mouse lines to identify and describe Dll1+ cells within mammary tumours.

The findings reported here are in my view confusing and, more importantly, the interpretation of the results is misleading and often incorrect. My major worries are:

- In Figure 2, the authors follow Dll1 expression using Dll1-mCherry mice and conclude that Dll1+ cells undergo a "cell fate change", instead of simply observing that in tumours, luminal cells also express Dll1. In these experiments they do not trace cells, they are just following the expression of Dll1. So, their conclusion is incorrect.

- Moreover, they analyse Dll1-expressing cells within mammary hyperplasia and tumours and clearly with tumour progression, the proportion of Dll1+ luminal (K8+) cells increases, whereas basal cells are strongly reduced, so the alleged "cell fate change" is simply the result of the fact that the total tumour cell population, including Dll1- cells, follows the same change in ratio from basal to luminal cells (see Fig 2d-e). Indeed, using the same mouse reporter line, the authors previously showed that, in the normal gland, a non-negligible percentage of Dll1+ cells were luminal (11.4%, as shown by FACS in Fig. 3b in Chakrabarti et al, Science 2018). Thus, the Dll1+ luminal cells found here in hyperplasia and tumours may very well derive from the amplification of a Dll1+/K8+ luminal cell population already present in the normal gland.

- It is absolutely necessary to re-interpret also the lineage tracing results presented in Fig. 3. Again, the conclusions are incorrect: if tracing is initiated in the hyperplasia or tumour stage, there is no true "cell fate change", as numerous luminal cells are already Dll1+ (Fig. 2d-e and Ext Data 2b-d). Thus, the initial luminal Dll1+ cells can easily generate the 23% luminal Tom+ cells found by FACS after 1 week of chase (Fig. 3c), in the absence of any cell fate change. The same reasoning holds true for tumours (compare Ext. Data 2d and Fig. 3 e-f).

- To reach the conclusions drawn by lineage tracing experiments, a single pulse of tamoxifen should be administered to mice (and not 3x) and tracing should be initiated before hyperplasia or tumours appear, in order to define if the tumour cells indeed derive from Dll1+ normal epithelial cells. The alleged increase in luminal cells traced, detected by FACS in Fig. 3c, cannot be interpreted in the absence of tracing in the normal mammary epithelium as a reference. Also, given the obvious change in basal to luminal ratio in hyperplasia and tumours, it is important to show the proportion of basal and luminal cells in all MEC in the FACS gates, and not just the Tomato+ cells.

- the Dll1- tumour cells isolated from PyMT-Dll1mCh tumours present broad mCherry signal upon treatment with doxorubicin (Extended Data Fig. 5f), implying that cells surviving chemotherapy start expressing Dll1 even when they had been sorted as Dll1- cells! If this is true, all experiments presented in Fig. 6 are not valid, as the differences observed in PyMT-Dll1-negative cells should not hold true upon chemotherapy treatment and conversion of Dll1- cells into Dll1+ cells! Conveniently, in Fig. 6e, only PyMT-Dll1+ cells are shown to indicate their increase in the presence of Doxorubicin.

Based on these major flaws, I am alarmed regarding how to interpret the rest of the presented data and would strongly recommend to the authors to re-evaluate their conclusions and perform a series of control experiments which may guide them towards a safer interpretation of their results.

I list here my other major concerns, in the hope that this could be of help to re-analyse the data obtained throughout the study.

- As briefly mentioned, in Fig. 2, there is no evidence for a change in cell fate specific to the Dll1+ cells, since the ratio of basal and luminal cells considerably changes in all cells of hyperplasia and tumours, compared to the normal mammary epithelium (Fig. 2d-e), see also Fig. 3d-f, where basically no K14+ cells are present, regardless what the authors state at page 6: "Confocal images further confirmed that the majority of tdTomato expression was in basal cells, which are K14+ and K8- (Fig. 3d)".

- The authors should assess the possibility that some cells may present co-expression of basal and luminal markers, as it has been reported in several instances in breast tumours. Also, given the paucity/nearly absence of K14+ basal cells in tumours, other basal markers such as K5, p63 or SMA should be tested. In this regard, K5 was unexpectedly found upregulated in Dll1+ tumour cells by RNAseq (Ext. Fig 4a), so it is definitely important to show K5 expression in these tumours by IF. Indeed, an increase in KRT5 expression might explain the better grafting efficiency and tumorsphere growth of Dll1+ tumour cells.

- Regarding the FACS gates in Fig. 3 c,e: the gating strategy is unusual; the authors should be more stringent in gating the Tomato+ population. If they are stricter in 3c, do they still see such high percentage of luminal cells (51.5%)? It is difficult to see a correlation between the cytometry data and the IF pictures.

- In Fig. 5b, the RNAseq profile of Dll1+ cells is enriched in the Mammary Stem Cell Up signature, which represents genes highly expressed in basal cells, defined as "Mammary stem cells" due to their high mammary repopulation capacity in transplantation assays (Lim et al. Transcriptome analyses of mouse and human mammary cell subpopulations reveal multiple conserved genes and pathways. Breast Cancer Res 12, 2010). Thus, this finding links the Dll1+ cells to their basal identity (like upregulation of K5 in Ext. Data 4a) and does NOT "highlight their tumour initiating role" as stated at page 13.

- Also related to Fig. 5: the RNAseq analysis shows an activation of NFkB in Dll1+ cells. Is there any sign of Notch pathway activation in these cells? Which pathways are enriched in Dll1- cells? Do Dll1- cells express Notch receptors, potentially receiving signals from adjacent Dll1+ cells? Lan et al. previously reported that in PyMT tumours, Shp2 induces Dll1 expression, resulting in inhibition of senescence. Do Dll1- cells show a senescence signature? These questions deserve to be at least discussed.

- As mentioned above, the results presented in Fig. 6 must be re-evaluated in light of the fact that doxorubicin induces Dll1 expression homogeneously in treated Dll1- tumour cells (Ext. Data 5f), hence it is not conceivable how the authors could detect differences between Dll1+ and Dll1- cells. Specifically, in Fig. 6b, the difference in tumour volume is difficult to appreciate given the large standard deviation of the control sample. Also, in Fig. 6e, the PyMT-Dll1- tumours should be analysed and split channels for red and green should be provided, as the potential co-localisation of the EdU and Dll1 signals is not possible to spot in the provided images. Data must be quantified. In Fig. 6f, the IF images of cleaved caspase-3 immunostaining need to be presented in the four samples.

- Fig. 7a-b: the sample of PyMT-Dll1+ tumours treated with doxorubicin alone in the same experimental setting has to be included and shown in 7a and 7b. Also, given the evidence presented in Ext. Data 5f, the effect of combination therapy on the PyMT-Dll1- tumours should also be assessed.

- In Fig. 7f-g: a double IF for NF-kB and Dll1 should be performed in both PyMT-Dll1+ and PyMT-Dll1- tumours. It is also surprising that the authors could find so compact tumours as the one

shown in Fig. 7f in Dll1- tumours treated with α Dll1+Dox, given the nearly absence of tumours in these conditions shown in Fig. 7b and the tumour volume of 0 mm³ in those mice (blue line in graph 7a)!

- In Fig. 7e, it is unclear if the pictures show metastases or inflamed lung nodules; could the authors perform a specific staining (i.e. pan-keratin?) to ensure the cells are indeed epithelial? Do those cells still express Dll1? Or mCherry?

- Fig. 8 points to a re-sensitization of the chemoresistant Dll1+ tumour cells by inhibition of NF κ B pathway. Those in vitro studies should be complemented by in vivo analyses to reinforce the model proposed in Fig. 8d.

- Finally, in the discussion, the authors say that Notch signalling has been involved in chemoresistance in TNBC (page 12), which is contradictory to the proposed role of Dll1, found by the authors to be upregulated in luminal breast cancer but not in TNBC. Furthermore, it is unclear which CSC would be targeted by GSI and which tumour types would show reduced growth upon GSI treatment? Page 12, no references are cited to support this statement.

- A more general remark: although the results obtained with the mouse models are potentially interesting from a clinical point of view, they would be considerably strengthened by in silico analysis of breast tumours gene expression in existing patient databases. For example, is there a correlation between Dll1 expression and the NF κ B signature in human luminal tumours?

Minor remarks:

- In Ext. Data 3, there is no quantification of immunofluorescence, so it is impossible to say that the levels of Dll1 are higher in lung nodules than in primary tumours.

- In Ext. Data 4c, how was the Venn diagram drawn to represent the Dll1-specific chromatin-accessible regions? Where are the chromatin-accessible regions in Dll1- tumour cells?

- In Fig. 5a: what is a "quiescence signature"? "Chang Cycling gene signature" is not self-explanatory...

Reviewer #2:

Remarks to the Author:

The manuscript by Kumar et al. convincingly demonstrates that Notch ligand DLL1 is upregulated during the process of hyperplasia and tumorigenesis in the MMTV-PyMT mouse model. DLL1 appears to be necessary for hyperplasia and transformation. Using informative animal models, the authors show that DLL1 is expressed in cells with a CSC phenotype and promotes doxorubicin resistance in part through NF- κ B, as well as metastasis. The experiments are generally rigorous and well-designed, and the data is compelling. The findings are innovative, in that a role of DLL1 in luminal breast cancer had not been described before. The findings are of potential translational relevance, since luminal breast cancers remain the most common and the largest cause of breast cancer mortality. However, a number of issues remain to be addressed:

1. Page 2: the statement describing the subtypes of human breast cancer is inaccurate and must be corrected. As a minimum, there are three main subtypes of human breast cancer (luminal, Her2-enriched and "triple negative" - not all of which are basal). Also, triple-negative breast cancer can and often does express Her2. However, it lacks genomic amplification of the ERBB2 locus and other contiguous genes, which produces massive overexpression of HER2 in HER2-enriched tumors. Furthermore, luminal tumors in humans have at least two molecular subtypes (luminal A and luminal B) with different mutational profiles, biological features and prognosis.
2. Figure 1 and extended data Figure 1: While there does appear to be a clear difference between

the PyMT and the MMTV-Wnt models, the difference may not be as clear-cut: in Extended Data Figure 1c, the survival analysis does trend towards a lower fraction of tumor-free animals in the DLL1-wt. The difference is non-significant but the sample size is very small. Thus, I don't think one can draw the definite conclusion that DLL1 plays no role in the MMTV-Wnt model or that the role of DLL1 is limited to luminal tumors (more on this later).

3. Page 7 and Figure 4: The higher tumorigenic ability of DLL1+ cells is clearly demonstrated. One question that remains, however, is whether DLL1+ and DLL1- cells cooperate. This has been demonstrated in other tumor models, where Notch signaling-high and -low cells form paracrine loops that promote tumorigenesis. Since in spontaneous tumors DLL1+ and DLL1- cells coexist, answering this question would strengthen the investigators' case.

4. Figure 5: while it is clear that the transcriptional profile of DLL1+ cells is different from that of DLL1- cells and that it is enriched in genes associated with metastasis and resistance to cell death, it is unclear whether any known Notch target genes are enriched in these cells. This addresses the important mechanistic questions whether DLL1+ cells activate Notch in contiguous DLL1+ cells, and whether the expression of DLL1 target genes is in fact Notch mediated. An alternative possibility is that DLL1+ cells activate Notch in DLL- cells, perhaps promoting their proliferation or survival. It is possible that DLL+ cells are actually Notch activity-low due to cis-inhibition, but activate Notch in DLL- cells. This should be explored.

5. An obvious, related question is which Notch paralogs are expressed in DLL1+ cells and whether they are activated. There is extensive literature on several Notch paralogs in breast CSC. Luminal CSC in human tumors appear to express at least 2 Notch paralogs (Notch1 and Notch4) and Notch4 expression has been proposed to be a feature of luminal breast CSC and required for their survival. These Notch paralogs may be expressed and active, expressed and inactive or not expressed in DLL1+ cells (but perhaps expressed in DLL- cells).

6. Page 12: The second sentence in the Discussion needs to be revised. There are multiple GSIs. GSI is a generic category of drugs which belong to different chemical series and are pharmacologically different. While several of them have been tested in clinical trials, none are FDA-approved.

7. Does expression of DLL1 correlate with recurrence-free or metastasis-free survival in human luminal A or luminal B tumors? This could be easily ascertained by analyzing public domain data and would greatly strengthen the conclusion of the manuscript that DLL1 drives a metastatic phenotype in luminal breast cancers.

8. What is the ER (ESR1) status of the DLL1+ cells? In humans, luminal tumors are ER-positive. In the PyMT model, early lesions tend to be ER-positive and advanced tumors tend to lose ER expression. In human tumors, not every cell is generally ER-positive, and ER-positive cells can support the growth of ER-negative cells through paracrine effects. Several scenarios are possible: 1. DLL1+ cells are ER+ and support the growth of other DLL1+ cells; 2. DLL1+ cells are ER+ and support the growth of ER- cells; DLL1- cells are ER+ and CSC are largely DLL+ etc. These scenarios are mechanistically relevant in that they would clarify the role of DLL1 in the promotion of luminal tumors.

Reviewer #1 (Remarks to the Author):

This work is based on a previous study from the same group (Chakrabarti et al, Science 2018), in which the authors used the Dll1-Cre mouse line as well as a Dll1-mCherry reporter mouse to characterise the Dll1⁺ cells in the normal mammary gland. Now the authors use the same mouse lines to identify and describe Dll1⁺ cells within mammary tumours. The findings reported here are in my view confusing and, more importantly, the interpretation of the results is misleading and often incorrect.

We are extremely sorry for the confusion and as per the reviewer's suggestion, we have made significant changes accordingly to improve the quality of the manuscript and data interpretation. We are grateful to the reviewer for reading our earlier article in Science, 2018. We previously used Dll1-GFP Cre and Dll1-mCherry reporter mouse models to characterize Dll1 protein in the normal mammary gland. In the present manuscript, we have generated transgenic mouse models by crossing them with MMTV-PyMT and MMTV-Wnt1 oncogenic background, in order to monitor the expression of Dll1 and to understand its role in breast cancer initiation and progression, which has never been performed in any earlier studies. Thus, our study is not a continuous study of the previous manuscript but asks novel questions such as status of Dll1, its spatial distribution and function in luminal breast cancer. In this manuscript, we have also explored the mechanistic basis behind Dll1 mediated phenotype in luminal cancer..

My major worries are:

- In Figure 2, the authors follow Dll1 expression using Dll1-mCherry mice and conclude that Dll1⁺ cells undergo a “cell fate change”, instead of simply observing that in tumours, luminal cells also express Dll1. In these experiments they do not trace cells, they are just following the expression of Dll1. So, their conclusion is incorrect.

We sincerely appreciate the comments made by the reviewer in the context of Fig. 2. However, we would like to clarify that we have shown lineage tracing experiments in Fig. 3c-d, where we found that progenitors of Dll1⁺ cells are primarily localized to the basal compartment in the hyperplastic mammary gland (Fig. 3c, d, pre-tumor stage) and in tumors, progenitors of Dll1⁺ cells are localized mostly in the luminal compartment (Fig. 3e, f). Nonetheless, considering the reviewer's suggestion, we have removed the term “cell fate change” from the manuscript.

- Moreover, they analyse Dll1-expressing cells within mammary hyperplasia and tumours and clearly with tumour progression, the proportion of Dll1⁺ luminal (K8⁺) cells increases, whereas basal cells are strongly reduced, so the alleged “cell fate change” is simply the result of the fact that the total tumour cell population, including Dll1⁻ cells, follows the same change in ratio from basal to luminal cells (see Fig 2d-e). Indeed, using the same mouse reporter line, the authors previously showed that, in the normal gland, a non-negligible percentage of Dll1⁺ cells were luminal (11.4%, as shown by FACS in Fig. 3b in Chakrabarti et al, Science 2018). Thus, the Dll1⁺ luminal cells found here in hyperplasia and tumours may very well derive from the amplification of a Dll1⁺/K8⁺ luminal cell population already present in the normal gland. It is absolutely necessary to re-interpret also the lineage tracing results presented in Fig.

We understand the reviewer's concern that the Dll1⁺ luminal cells found here in hyperplasia and tumors may have been derived from amplification of a small Dll1⁺/K8⁺ luminal cell population already present in the normal gland. However, when we examine Dll1 expression in MMTV-Wnt1 hyperplasia and tumors (Wnt1; Dll1^{mCherry}), we do not observe a similar increase in Dll1⁺K8⁺ population as shown below in Figure R1. MMTV-Wnt1 tumors have mixed population of basal and luminal cells, however, Dll1 expression is restricted to basal population. Thus, it is possible that the Dll1⁺ luminal cells in MMTV-PyMT tumor may have been derived from basal population. Nonetheless, since we do not have lineage tracing from very early stage like the way reviewer suggested, we have removed the term "cell fate change" from the manuscript. The highlight of the manuscript is determining the molecular mechanism of Dll1 on tumor cells survival after chemoresistance by NF-κβ signalling, which is where we focussed for the revision.

Figure R1. Flow cytometry data show total epithelial cell population in normal mammary gland, hyperplasia and in tumor stage from Wnt1-Dll1^{mCherry} mice. A,B Flow cytometry profiles show basal and luminal population of Lin⁻ cells (A) or Lin⁻ Dll1⁺ cells (B, mCherry⁺) during luminal tumor progression. As hyperplasia progresses to tumor, Dll1⁺ cells mostly stay in basal population.

3. Again, the conclusions are incorrect: if tracing is initiated in the hyperplasia or tumour stage, there is no true "cell fate change", as numerous luminal cells are already Dll1⁺ (Fig. 2d-e and Ext Data 2b-d). Thus, the initial luminal Dll1⁺ cells can easily generate the 23% luminal Tom⁺ cells found by FACS after 1 week of chase (Fig. 3c), in the absence of any cell fate change. The same reasoning holds true for tumours (compare Ext. Data 2d and Fig. 3 e-f). - To reach the conclusions drawn by lineage tracing experiments, a single pulse of tamoxifen should be administered to mice (and not 3x) and tracing should be initiated before hyperplasia or tumours appear, in order to define if the tumour cells indeed derive from Dll1⁺ normal epithelial cells.

The alleged increase in luminal cells traced, detected by FACS in Fig. 3c, cannot be interpreted in the absence of tracing in the normal mammary epithelium as a reference. Also, given the obvious change in basal to luminal ratio in hyperplasia and tumours, it is important to show the proportion of basal and luminal cells in all MEC in the FACS gates, and not just the Tomato+ cells.

We appreciate the suggestions made by the reviewer regarding the lineage tracing experiment. However, to perform the suggested experiment, triple transgenic mice (PyMT-Dll1GFP; tdTomato) need to be induced for cell lineage tracing at the age of 3-4 weeks. Due to the COVID-19 issues of schools closing, we could not breed mice to get that piece of data as it would have taken additional 3-4 months. Furthermore, we would like to emphasize that the revised manuscript does not highlight the “cell fate change” of Dll1 in mammary hyperplasia or tumor and we have removed that term from the manuscript. Rather, understanding the molecular mechanism of Dll1 mediated chemoresistance via NF- κ B pathway is the main focus of the manuscript. In the revised manuscript, we have now added *in vivo* mice data demonstrating the involvement of NF- κ B in Dll1 mediated chemoresistance to make our claim stronger than before. Also, *in silico* data analysis using human METABRIC dataset highlights the clinical significance of our study and also shows a strong correlation between *DLL1* and *NF- κ B1* in human luminal A and B breast cancer patient samples.

We have added the basal and luminal gates in all MECs, as suggested by the reviewer in new Supplementary Figure 3a, b in the revised manuscript.

- the Dll1- tumour cells isolated from PyMT-Dll1mCh tumours present broad mCherry signal upon treatment with doxorubicin (Extended Data Fig. 5f), implying that cells surviving chemotherapy start expressing Dll1 even when they had been sorted as Dll1- cells! If this is true, all experiments presented in Fig. 6 are not valid, as the differences observed in PyMT-Dll1- negative cells should not hold true upon chemotherapy treatment and conversion of Dll1- cells into Dll1+ cells! Conveniently, in Fig. 6e, only PyMT-Dll1+ cells are shown to indicate their increase in the presence of Doxorubicin.

We appreciate the reviewer’s concern. It is true that in our PyMT-Dll1^{mcherry} model, we found that even though Dll1⁻ tumor cells initially have no mcherry (Dll1) expression, but after transplantation, some tumor cells re-express mcherry (Dll1) expression. However, that level of expression is significantly lower in Dll1⁻ tumors compared to Dll1⁺ tumor cells. We have replaced old images with revised new ones which clarifies this point. Please see the revised Fig. 6g and h and supplementary figures 8c, d. It is true that after chemotherapy, the levels of Dll1⁺ cells increase in Dll1⁻ tumors; however, notably, the number of Dll1⁺ cells in Dll1⁻ tumors is significantly higher compared to Dll1⁻ tumors. We have now added Dll1⁺ and Dll1⁻ tumors stained with mCherry antibody to show the relative distribution of Dll1 in these two tumor types with and without chemotherapy (Fig. 6g and h and supplementary figures 8c, d).

Based on these major flaws, I am alarmed regarding how to interpret the rest of the presented data and would strongly recommend to the authors to re-evaluate their conclusions and perform a series of control experiments which may guide them towards a safer interpretation of their results.

I list here my other major concerns, in the hope that this could be of help to re-analyse the data obtained throughout the study.

We highly appreciate the suggestions made by the reviewer. Please find below our responses to these comments.

- As briefly mentioned, in Fig. 2, there is no evidence for a change in cell fate specific to the Dll1⁺ cells, since the ratio of basal and luminal cells considerably changes in all cells of hyperplasia and tumours, compared to the normal mammary epithelium (Fig. 2d-e), see also Fig. 3d-f, where basically no K14⁺ cells are present, regardless what the authors state at page 6: “Confocal images further confirmed that the majority of tdTomato expression was in basal cells, which are K14⁺ and K8⁻ (Fig. 3d)”.

Again, we apologize for our misinterpretation, and thus we removed the term “cell fate change” from the revised manuscript due to lack of adequate lineage tracing experiment. Please see response above. In the revised manuscript, instead of “cell fate change” we have described them as Dll1⁺ basal or Dll1⁺ luminal cells.

Our original zoomed-in image in Fig. 3d does not show the correct proportion of K14⁺ cells; therefore, we have replaced Fig. 3d (left and right panel) showing the presence of fewer K14⁺ cells compared to K8⁺ cells. We have modified the statement accordingly in the revised manuscript. From both FACS and IF (location) data, it is clear at hyperplasia stage the tomato expressing cells are more in the basal position, however in tumor context we do see Dll1⁺ cells colocalize with K8⁺ luminal cells. Please see the response below.

- The authors should assess the possibility that some cells may present co-expression of basal and luminal markers, as it has been reported in several instances in breast tumours. Also, given the paucity/nearly absence of K14⁺ basal cells in tumours, other basal markers such as K5, p63 or SMA should be tested. In this regard, K5 was unexpectedly found upregulated in Dll1⁺ tumour cells by RNAseq (Ext. Fig 4a), so it is definitely important to show K5 expression in these tumours by IF. Indeed, an increase in KRT5 expression might explain the better grafting efficiency and tumorsphere growth of Dll1⁺ tumour cells.

We would like to highlight that there is a reasonable amount of K14⁺ cells in the tumors along with very high number of K8⁺ luminal cells. We apologize for choosing the wrong area. We have now replaced Fig. 3d (left and right panel) in the revised manuscript.

- Regarding the FACS gates in Fig. 3 c,e: the gating strategy is unusual; the authors should be more stringent in gating the Tomato⁺ population. If they are stricter in 3c, do they still see such high percentage of luminal cells (51.5%)? It is difficult to see a correlation between the cytometry data and the IF pictures.

We have now applied stricter gates for tomato in FACS. We found that in the hyperplastic mammary gland, higher expression of Tomato is localized in the basal compartment, while those cells with low Tomato expression localize in the luminal compartment (Figure R2). We have provided images with stringent gating of Tomato⁺ cells for review purpose, please see below

Figure R2. From our revised manuscript images, we now observe a good correlation in FACS and IF data showing expression of Dll1 in the basal and luminal compartment in both hyperplasia and in tumor stage.

Figure R2. The representative flow profiles of epithelial tumor cells ($CD45^+CD31^-TER119^-$) from hyperplasia and tumors from PyMT; $Dll1-GFP-Cre$; $tdTomato$ mice show $tdTomato^+$ cells in luminal ($CD24^{high}CD29^{low}$) and basal ($CD24^{low}CD29^{high}$) population at hyperplasia (upper panel) or tumor stage (lower panel).

- In Fig. 5b, the RNAseq profile of $Dll1^+$ cells is enriched in the Mammary Stem Cell Up signature, which represents genes highly expressed in basal cells, defined as “Mammary stem cells” due to their high mammary repopulation capacity in transplantation assays (Lim et al. Transcriptome analyses of mouse and human mammary cell subpopulations reveal multiple conserved genes and pathways. Breast Cancer Res 12, 2010). Thus, this finding links the $Dll1^+$ cells to their basal identity (like upregulation of K5 in Ext. Data 4a) and does NOT “highlight their tumour initiating role” as stated at page 13.

We stated the tumor-initiating function of $Dll1$ based on Figure 4b, c. In page 13 (revised page 8), we have modified that sentence to describe clearly the reason for mentioning the tumor-initiating function of $Dll1$.

- Also related to Fig. 5: the RNAseq analysis shows an activation of $NF-\kappa\beta$ in $Dll1^+$ cells. Is there any sign of Notch pathway activation in these cells? Which pathways are enriched in $Dll1^+$ cells? Do $Dll1^-$ cells express Notch receptors, potentially receiving signals from adjacent $Dll1^+$ cells? Lan et al. previously reported that in PyMT tumours, Shp2 induces $Dll1$ expression, resulting in inhibition of senescence. Do $Dll1^-$ cells show a senescence signature? These questions deserve to be at least discussed.

The juxtacrine aspect of Notch signaling is indeed a very interesting question raised by the reviewer. RNA-seq shows that $Dll1^+$ tumor cells have enriched signature for Notch pathway genes suggesting that $Dll1^+$ tumor cells may be talking to other $Dll1^+$ tumor cells (new Supplementary Fig 5d). To explore this possibility further, we sorted $Dll1^+$ and $Dll1^-$ tumor cells from PyMT- $Dll1^{mCherry}$ tumors and quantified the mRNA levels of Notch receptors (*Notch 1-4*) and the Notch-signaling dependent gene, *Hes1* and *Hey1*. We observed Notch receptor *N1* and

N4 was higher in *Dll1*⁺ cells, whereas Notch receptor *N3* was significantly higher in *Dll1*⁻ tumor cells, suggesting that crosstalk can happen between *Dll1*⁺ tumor cells through Notch receptor 1 and 4.

Furthermore, *Hey1* was increased in *Dll1*⁺ tumor cells, suggesting Notch signalling is activated in *Dll1*⁺ tumor cells. Future comprehensive studies will delineate the mechanism of Notch signaling between different tumor cell populations such as *Dll1*⁺ and *Dll1*⁻ cell population. Please see Figure R3 below.

Figure R3. Bar graph shows the levels of indicated mRNAs in *Dll1*⁺ (*mCherry*⁺) and *Dll1*⁻ (*mCherry*⁻) tumor cells isolated from spontaneous *PyMT-Dll1*^{*mCherry*} tumors.

Additionally, to explore the role of *Dll1* in senescence, we have analysed 1140 luminal patients using METABRIC database^{1,2}.

Interestingly, there were 3 gene sets for senescence enriched in *DLL1*^{high} Luminal A and B patients and 4 gene sets for senescence were enriched for *DLL1*^{low} Luminal A and B patients, suggesting no significant difference in senescence signatures between the two subgroup of patients (*DLL1*^{low} or *DLL1*^{high}) patients. We have provided this data (Please see Figure R4) below for review purpose.

senescence enriched in *DLL1*^{high} Luminal A and B patients and 4 gene sets for senescence were enriched for *DLL1*^{low} Luminal A and B patients, suggesting no significant difference in senescence signatures between the two subgroup of patients (*DLL1*^{low} or *DLL1*^{high}) patients. We

Figure R4. Gene set enrichment analyses (GSEA) demonstrate Tang senescence TP53 target gene sets in *Dll1*^{high} and *Dll1*^{low} patient tumors (n=1140). METABRIC dataset was used to extract this data^{1,2}.

- As mentioned above, the results presented in Fig. 6 must be re-evaluated in light of the fact that doxorubicin induces *Dll1* expression homogeneously in treated *Dll1*⁻ tumour cells (Ext. Data 5f), hence it is not conceivable how the authors could detect differences between *Dll1*⁺ and *Dll1*⁻ cells. Specifically, in Fig. 6b, the difference in tumour volume is difficult to appreciate given the large standard deviation of the control sample. Also, in Fig. 6e, the *PyMT-Dll1*⁻ tumours should be analysed and split channels for red and green should be provided, as the potential co-localisation of the EdU and *Dll1* signals is not possible to spot in the provided images. Data must be quantified. In Fig. 6f, the IF images of cleaved caspase-3 immunostaining need to be presented in the four samples.

As explained in the above responses, we would like to mention that even though Dll1⁻ tumor cells gain Dll1 expression after chemoresistance, the expression is not homogenous like the Dll1⁺ tumor cells. Please see the revised images in Fig. 6g and h and supplementary figures 8c, d. In Fig. 6b, the Dll1⁺ tumors grow slow so the size of tumors is small compared to the Dll1⁻ tumors (please see the y-axis of Figs. 6b-c). Also, Dll1⁺ tumors are chemoresistant therefore, the size between control and dox groups is not statistically different. Hence the error bars are slightly larger in the control group.

As per the reviewer's suggestion, we have provided split channel images for Fig. 6e, and quantified the data in Supplementary Fig. 7c. Since cleaved caspase-3 antibody did not work for IF, we performed TUNEL assay using PyMT-Dll1⁺ and PyMT-Dll1⁻ tumors after doxorubicin treatment. The TUNEL staining showed a higher number of apoptotic cells among Dll1⁻ cells, suggesting chemoresistant nature of Dll1⁺ tumor cells (Fig. 6g, h). Notably, Dll1⁻ tumor cells show more TUNEL⁺ cells after chemotherapy compared to Dll1⁺ tumor cells (Fig. 6g, h).

- Fig. 7a-b: the sample of PyMT-Dll1⁺ tumours treated with doxorubicin alone in the same experimental setting has to be included and shown in 7a and 7b. Also, given the evidence presented in Ext. Data 5f, the effect of combination therapy on the PyMT-Dll1⁻ tumours should also be assessed.

As per the suggestion of the reviewer to include doxorubicin (Dox) alone in the same experiment, we performed a new experiment with control, α Dll1-antibody, Dox and both α Dll1-ab + Dox treatment groups. In our revised manuscript, the combination of both α Dll1-ab and Dox inhibited tumor growth and metastasis (Fig 7a-e). We have added new data in the revised manuscript.

In Supplementary Fig. 5 (new Supplementary Fig 7a, b and Fig. 6b, c), we assessed chemoresistant characteristics of Dll1⁺ or Dll1⁻ tumors to build the foundation for Dll1-blocking and combination treatments. It is now clear from Fig. 6-e-h, that Dll1⁻ cells are sensitive to chemotherapy compared to Dll1⁺ cells. The focus of the paper is on the Dll1⁺ cells, so we did not pursue the combination therapy for Dll1⁻ tumor cells.

- In Fig. 7f-g: a double IF for NF- κ B and Dll1 should be performed in both PyMT-Dll1⁺ and PyMT-Dll1⁻ tumours. It is also surprising that the authors could find so compact tumours as the one shown in Fig. 7f in Dll1⁻ tumours treated with α Dll1+Dox, given the nearly absence of tumours in these conditions shown in Fig. 7b and the tumour volume of 0 mm³ in those mice (blue line in graph 7a)!

We performed an IHC staining for NF- κ β on PyMT-Dll1⁺ and PyMT-Dll1⁻ tumors. As expected, the Dll1⁺ tumors showed higher nuclear NF- κ β staining compared with Dll1⁻ tumors. We have provided the respective images in the revised manuscript (Fig. 5h-i).

We apologize for the images shown earlier in Fig. 7f. We have now repeated the experiments and found that indeed the tumors treated with combination (Dll1 antibody + doxorubicin) therapy or Dll1 antibody are smaller compared to control, but tumors are present in small size. Please see revised Fig. 7a, b. We performed IHC staining for NF- κ β and found that NF- κ β ⁺ cells

were fewer and more cytoplasmic in the alone DLL1 antibody and combination-treated tumors compared to control and doxorubicin (Fig. 7f, g).

- In Fig. 7e, it is unclear if the pictures show metastases or inflamed lung nodules; could the authors perform a specific staining (i.e. pan-keratin?) to ensure the cells are indeed epithelial? Do those cells still express Dll1? Or mCherry?

The metastatic lung nodules showed substantive mCherry (Dll1) expression in Fig 7c under the dissection microscope, which led us to make sure about the presence of lung nodules. We have replaced Fig. 7e with better H&E images in the revised manuscript (please see new Supplementary Fig 9b). Following the reviewer's suggestion, we have stained these lung nodules with mCherry (Dll1), K14 and K8 antibodies to confirm the presence of Dll1⁺ tumor cells in the metastatic nodules (Fig. 7e and Supplementary Fig. 9b). Furthermore, all lung H&E images were evaluated by a pathologist, Dr. Andres J.P Klein-Szanto, Fox Chase Cancer Center, who also confirmed the lung images to have metastatic cells from a histopathological point of view.

- Fig. 8 points to a re-sensitization of the chemoresistant Dll1⁺ tumour cells by inhibition of NF- κ B pathway. Those in vitro studies should be complemented by in vivo analyses to reinforce the model proposed in Fig. 8d.

As per the reviewer's suggestion, we treated PyMT-Dll1⁺ tumors to various combinations of α Dll1-antibody, doxorubicin and IMD, an inhibitor of NF- κ B³⁻⁵. We found that tumor growth was significantly slower in α Dll1-ab + IMD or α Dll1-ab + Dox combination compared to only doxorubicin treatment group, suggesting that Dll1 is essential for these tumor growths and progression (Fig. 8d,e). In addition, the growth of triple treatment (α Dll1-ab + doxorubicin + IMD) tumors were significantly slower than α Dll1-ab + IMD or α Dll1-ab + Dox suggesting further benefit of triple drug treatments (Fig. 8d,e). Tumor growth in mice treated with Doxorubicin alone was considered for comparison amongst different groups.

Note: Due to lockdown amid COVID-19 pandemic, we had to minimize the animal numbers per experiment, therefore, we used the doxorubicin alone data for two panels in Fig. 7a and 8d. Also, we had to terminate the experiment 10 days earlier than usual, therefore the size of tumors are a bit smaller than earlier experiment.

- Finally, in the discussion, the authors say that Notch signalling has been involved in chemoresistance in TNBC (page 12), which is contradictory to the proposed role of Dll1, found by the authors to be upregulated in luminal breast cancer but not in TNBC. Furthermore, it is unclear which CSC would be targeted by GSI and which tumour types would show reduced growth upon GSI treatment? Page 12, no references are cited to support this statement.

We understand the concern raised by the reviewer in this aspect. In old page 12 (revise manuscript page 13), where we mentioned the studies which demonstrate Notch signalling and TNBC mostly focusses on Notch receptors and Jagged1/2 in TNBC and not Dll1. However, the function of Dll1 is more evident in luminal cancer compared to basal based on Figure 1 and Supplementary Figure 1a-c of this manuscript and in our published work⁶⁻¹¹. Therefore, we don't believe that the data is contradictory here in this context. About the GSI targeting CSC, we

have modified the respective sentence on page 12 (revised manuscript page 13) and added relevant cited articles.

- A more general remark: although the results obtained with the mouse models are potentially interesting from a clinical point of view, they would be considerably strengthened by in silico analysis of breast tumours gene expression in existing patient databases. For example, is there a correlation between Dll1 expression and the NF- κ B signature in human luminal tumours?

This is a great suggestion. We analyzed METABRIC dataset and found a strong correlation between *DLL1* expression and *NF- κ B* signatures. We used the mouse Dll1 signatures from both GFP and mCherry reporter models and checked for correlation to *DLL1*^{high} patients and found that mouse Dll1^{high} signature strongly correlates to human patient tumors with high *DLL1* expression (Fig. 5j and Supplementary Fig. 6a). METABRIC dataset comprises gene expressions of around 1100 patient samples for luminal A and B subtypes^{1,2}. Interestingly, *DLL1*^{high} patient tumors show activated NF- κ B signaling and *NF- κ B* gene shows a strong positive correlation with *DLL1* expression (Fig. 5k-m, and Supplementary Fig. 6b-d).

Minor remarks:

- In Ext. Data 3, there is no quantification of immunofluorescence, so it is impossible to say that the levels of Dll1 are higher in lung nodules than in primary tumours.

We are thankful to the reviewer for highlighting this issue. We calculated the H-score by quantifying both intensity of Dll1 and abundance of Dll1⁺ cells in more than three tumors and lung nodules. The quantification data have been provided in Supplementary Fig. 4e-f. Note: The H-score value is the product of abundance of cells expressing respective protein (scale of 0–100) multiplied by the intensity of expression of that protein (scale of 0–3).

- In Ext. Data 4c, how was the Venn diagram drawn to represent the Dll1-specific chromatin-accessible regions? Where are the chromatin-accessible regions in Dll1- tumour cells?

We are thankful to the reviewer for highlighting this issue. The older pie chart shows differential peaks of Dll1⁺ and Dll1⁻ tumor cells and not just Dll1-specific chromatin-accessible regions. We have now generated pie charts for Dll1⁺ and Dll1⁻ tumor samples and found no profound change in the differential peaks of Dll1⁺ and Dll1⁻ tumors compared to each other. We have corrected this part in the revised manuscript (Supplementary Fig. 5f).

- In Fig. 5a: what is a “quiescence signature”? “Chang Cycling gene signature” is not self-explanatory...

Acknowledging the reviewer’s suggestion, we have elaborated the details of quiescence signatures in more detail in the revised manuscript. We would like to thank the reviewer for careful and thorough reading of our manuscript and appreciate his critical comments and suggestions. The reviewer’s comments were elaborate and nicely written, which guided us to thoroughly revise the manuscript. We believe that new data added in the revised manuscript will bring more clarity and have significantly improved the quality of the manuscript.

Reviewer #2 (Remarks to the Author):

The manuscript by Kumar et al. convincingly demonstrates that Notch ligand DLL1 is upregulated during the process of hyperplasia and tumorigenesis in the MMTV-PyMT mouse model. DLL1 appears to be necessary for hyperplasia and transformation. Using informative animal models, the authors show that DLL1 is expressed in cells with a CSC phenotype and promotes doxorubicin resistance in part through NF- κ B, as well as metastasis. The experiments are generally rigorous and well-designed, and the data is compelling. The findings are innovative, in that a role of DLL1 in luminal breast cancer had not been described before. The findings are of potential translational relevance, since luminal breast cancers remain the most common and the largest cause of breast cancer mortality. However, a number of issues remain to be addressed:

We thank the reviewer for careful reading and understanding our manuscript, and for positive comments. We appraise his/her thoughtful and constructive suggestions to improve the quality of manuscript.

1. Page 2: the statement describing the subtypes of human breast cancer is inaccurate and must be corrected. As a minimum, there are three main subtypes of human breast cancer (luminal, Her2-enriched and “triple negative” - not all of which are basal). Also, triple-negative breast cancer can and often does express Her2. However, it lacks genomic amplification of the ERBB2 locus and other contiguous genes, which produces massive overexpression of HER2 in HER2-enriched tumors. Furthermore, luminal tumors in humans have at least two molecular subtypes (luminal A and luminal B) with different mutational profiles, biological features and prognosis.

We appreciate the statement made by the reviewer and have modified the introduction part of the revised manuscript as per the reviewer’s suggestion.

2. Figure 1 and extended data Figure 1: While there does appear to be a clear difference between the PyMT and the MMTV-Wnt models, the difference may not be as clear-cut: in Extended Data Figure 1c, the survival analysis does trend towards a lower fraction of tumor-free animals in the DLL1-wt. The difference is non-significant but the sample size is very small. Thus, I don’t think one can draw the definite conclusion that DLL1 plays no role in the MMTV-Wnt model or that the role of DLL1 is limited to luminal tumors (more on this later).

The reviewer has raised a valid point. Following the reviewer’s suggestion, we added more number of mice in Wnt-Dll1^{ckO} (n=22) and Wnt-Dll1^{WT} (n=10) groups; however, the data is still not significant. We have accordingly modified the data and discussion in the revised manuscript (Supplementary Fig. 1c).

3. Page 7 and Figure 4: The higher tumorigenic ability of DLL1+ cells is clearly demonstrated. One question that remains, however, is whether DLL1+ and DLL1- cells cooperate. This has been demonstrated in other tumor models, where Notch signaling-high and -low cells form

paracrine loops that promote tumorigenesis. Since in spontaneous tumors DLL1⁺ and DLL1⁻ cells coexist, answering this question would strengthen the investigators' case.

The juxtacrine aspect of Notch signaling is indeed a very interesting question raised by the reviewer. RNA-seq shows that Dll1⁺ tumor cells have enriched signature for Notch pathway genes suggesting that Dll1⁺ tumor cells may be talking to other Dll1⁺ tumor cells (new Supplementary Fig 5d). To explore this possibility further, we sorted Dll1⁺ and Dll1⁻ tumor cells from PyMT-Dll1^{mCherry} tumors. Further, we quantified the mRNA levels of Notch receptors (*Notch 1-4*) and Notch-signaling dependent gene, *Hes1* and *Hey1*. We observed Notch receptor *N1* and *N4* was higher in Dll1⁺ cells, whereas Notch receptor *N3* was significantly higher in Dll1⁻ tumor cells, suggesting that crosstalk can happen between Dll1⁺ cells through Notch receptor 1 and 4. Furthermore, *Hey1* was increased in Dll1⁺ tumor cells, suggesting Notch signalling is activated in Dll1⁺ tumor cells. Future comprehensive studies will delineate the mechanism of Notch signaling between different tumor cell populations such as Dll1⁺ and Dll1⁻ cell population. Please see Figure R3 below.

Figure R3. Bar graph shows the levels of indicated mRNAs in Dll1⁺ (mCherry⁺) and Dll1⁻ (mCherry⁻) tumor cells isolated from spontaneous PyMT-Dll1^{mCherry} tumors.

4. Figure 5: while it is clear that the transcriptional profile of DLL1⁺ cells is different from that of DLL1⁻ cells and that it is enriched in genes associated with metastasis and resistance to cell death, it is unclear whether any known Notch target genes are enriched in these cells. This addresses the important mechanistic questions whether DLL1⁺ cells activate Notch in contiguous DLL1⁺ cells, and whether the expression of DLL1 target genes is in fact Notch mediated. An alternative possibility is that DLL1⁺ cells activate Notch in DLL⁻ cells, perhaps promoting their proliferation or survival. It is possible that DLL⁺ cells are actually Notch activity-low due to cis-inhibition, but activate Notch in DLL⁻ cells. This should be explored.

This is a great suggestion. Please see the response to Reviewer 1 and associated image Figure R3. In brief, we found that Notch signalling is activated in Dll1⁺ tumor cells, which needs further studies in the future elucidating mechanism of crosstalk between Dll1⁺ tumor cells in luminal tumors.

5. An obvious, related question is which Notch paralogs are expressed in DLL1⁺ cells and whether they are activated. There is extensive literature on several Notch paralogs in breast CSC. Luminal CSC in human tumors appear to express at least 2 Notch paralogs (Notch1 and Notch4) and Notch4 expression has been proposed to be a feature of luminal breast CSC and required for their survival. These Notch paralogs may be expressed and active, expressed and inactive or not expressed in DLL1⁺ cells (but perhaps expressed in DLL⁻ cells).

We are thankful for this comment on Notch-signaling crosstalk. Please see the response above.

6. Page 12: The second sentence in the Discussion needs to be revised. There are multiple GSIs. GSI is a generic category of drugs which belong to different chemical series and are pharmacologically different. While several of them have been tested in clinical trials, none are FDA-approved.

We are thankful to the reviewer for providing us this information on GSI. We have removed the sentence (Revised manuscript page no 13)

7. Does expression of DLL1 correlate with recurrence-free or metastasis-free survival in human luminal A or luminal B tumors? This could be easily ascertained by analyzing public domain data and would greatly strengthen the conclusion of the manuscript that DLL1 drives a metastatic phenotype in luminal breast cancers.

In our previous study, we have analysed the role of DLL1 in metastasis in luminal tumors by using KM plotter database. We found that higher expression of DLL1 leads to the poor prognosis in luminal A and B subtypes (Supplementary Fig. S1 B-C, Kumar, et al., Oncogene 2019)¹².

8. What is the ER (ESR1) status of the DLL1+ cells? In humans, luminal tumors are ER-positive. In the PyMT model, early lesions tend to be ER-positive and advanced tumors tend to lose ER expression. In human tumors, not every cell is generally ER-positive, and ER-positive cells can support the growth of ER-negative cells through paracrine effects. Several scenarios are possible: 1. DLL1+ cells are ER+ and support the growth of other DLL1+ cells; 2. DLL1+ cells are ER+ and support the growth of ER- cells; DLL1- cells are ER+ and CSC are largely DLL+ etc. These scenarios are mechanistically relevant in that they would clarify the role of DLL1 in the promotion of luminal tumors.

This is a fascinating question raised by the reviewer. To test this, we isolated the Dll1^{+/-} tumor cells from PyMT-Dll1^{mCh} tumors by cell sorting. Then, we quantified the expression levels of *Era* gene in Dll1⁺ or Dll1⁻ tumor cells. We found no major difference in *Era* expression between Dll1⁺ and Dll1⁻ tumors. Our previous report suggests that Estrogen signaling stabilizes the protein levels of DLL1 in luminal tumors¹². So, it is possible that in MMTV-PyMT tumors, similar mechanisms exist in Dll1⁺ tumor cells, but in Dll1⁻ tumor cells, ER α may have an additional function, which needs future evaluation.

Figure R5. Bar graphs show the levels of *Dll1* and *ER α* (*ESR1*) mRNAs in Dll1⁺ and Dll1⁻ tumor cells isolated from spontaneous PyMT-Dll1^{mCh} tumors.

- 1 Curtis, C. *et al.* The genomic and transcriptomic architecture of 2,000 breast tumours reveals novel subgroups. *Nature* **486**, 346-352, doi:10.1038/nature10983 (2012).
- 2 Pereira, B. *et al.* The somatic mutation profiles of 2,433 breast cancers refines their genomic and transcriptomic landscapes. *Nature communications* **7**, 11479, doi:10.1038/ncomms11479 (2016).
- 3 Farr, G. W. *et al.* Functionalized Phenylbenzamides Inhibit Aquaporin-4 Reducing Cerebral Edema and Improving Outcome in Two Models of CNS Injury. *Neuroscience* **404**, 484-498, doi:10.1016/j.neuroscience.2019.01.034 (2019).
- 4 Zhang, J. X. *et al.* LINC01410-miR-532-NCF2-NF-kB feedback loop promotes gastric cancer angiogenesis and metastasis. *Oncogene* **37**, 2660-2675, doi:10.1038/s41388-018-0162-y (2018).
- 5 Zwicker, S. *et al.* Interleukin 34: a new modulator of human and experimental inflammatory bowel disease. *Clin Sci (Lond)* **129**, 281-290, doi:10.1042/CS20150176 (2015).
- 6 Venkatesh, V. *et al.* Targeting Notch signalling pathway of cancer stem cells. *Stem Cell Investig* **5**, 5, doi:10.21037/sci.2018.02.02 (2018).
- 7 Lamy, M. *et al.* Notch-out for breast cancer therapies. *N Biotechnol* **39**, 215-221, doi:10.1016/j.nbt.2017.08.004 (2017).
- 8 Zhu, H. *et al.* Elevated expression of notch1 is associated with metastasis of human malignancies. *Int J Surg Pathol* **21**, 449-454, doi:10.1177/1066896913496146 (2013).
- 9 Giuli, M. V., Giuliani, E., Screpanti, I., Bellavia, D. & Checquolo, S. Notch Signaling Activation as a Hallmark for Triple-Negative Breast Cancer Subtype. *Journal of oncology* **2019**, 8707053, doi:10.1155/2019/8707053 (2019).
- 10 Nagamatsu, I. *et al.* NOTCH4 is a potential therapeutic target for triple-negative breast cancer. *Anticancer research* **34**, 69-80 (2014).
- 11 Speiser, J. J., Ersahin, C. & Osipo, C. The functional role of Notch signaling in triple-negative breast cancer. *Vitamins and hormones* **93**, 277-306, doi:10.1016/B978-0-12-416673-8.00013-7 (2013).
- 12 Kumar, S. *et al.* Estrogen-dependent DLL1-mediated Notch signaling promotes luminal breast cancer. *Oncogene* **38**, 2092-2107, doi:10.1038/s41388-018-0562-z (2019).

Reviewers' Comments:

Reviewer #1:

Remarks to the Author:

The revised manuscript has corrected some of the problems existing in the original work, but unfortunately the authors have failed to address several major concerns that were raised by reviewers. Suggested experiments to improve or clarify the results were in most cases not performed or disregarded and the authors insist on drawing strong conclusions that, in my view, are not supported by solid experimental evidence.

The most critical concern in my opinion is related to lineage tracing analysis presented in Figure 3, which I advise the authors to remove from the study, as it is misinterpreted and will generate unnecessary confusion. Indeed, it is not just the term "cell fate change" that I was criticising, but the idea that Dll1+ basal cells would give rise to luminal cells within tumours. Clearly PyMT tumours, unlike the MMTV-Wnt1 tumours (see also Fig. R1), present a major shift in the basal to luminal ratio (now shown in Fig. S3), which is reflected by the expansion of the luminal cells that express Dll1. This does NOT mean that these cells derive from basal cells and in fact in the new FACS gating strategy in Fig. R2, one can appreciate that many Tomato+ cells are found in the stromal fraction (29.1%!). This does not mean that these stromal cells derive from Dll1+ basal cells, so why should the authors conclude that it is the case for the increased Dll1+ luminal cell fraction? In the legend to Figure 2 we read: "Fig. 2: Dll1+ basal cells change to K8+ luminal cells during PyMT tumor development". This is not correct. In this Figure, they use a reporter mouse that follows Dll1 expression, they DO NOT trace Dll1-expressing cells. The data simply showed the shift of Dll1 expression from basal cells in physiological conditions to luminal cells K8+ in tumors.

In the rebuttal the authors claim that "The highlight of the manuscript is determining the molecular mechanism of Dll1 on tumor cells survival after chemoresistance by NF- κ B signaling, which is where we focused for the revision"; this is the main argument to explain why they avoided to address my question regarding whether luminal Dll1-expressing cells are, indeed, the cells of origin of luminal tumors with lack of evidence for a cell fate change. If that is their final argument, it is even more justifiable to remove the entire Figure 3 and clarify the message. IF for K5, p63 or SMA, as I had requested, is absolutely required to explore the possibility of double positive cells and to explain the "disappearance" of K14+ cells in PyMT tumours. Moreover, in Figure 3, an increase from 11 to 14.9% in the percentage of Tom+ cells at hyperplasia stage and advanced tumor stage is reported; however, the images are not representative of such percentages.

In several instances, I found it troubling that the authors acknowledged my remarks and decided to replace specific images with other ones that better agree with the conclusions they have reached. How do we know which images are representative of the results, the originally submitted ones or the revised ones?

One striking example relates to my observation that cells surviving chemotherapy start expressing Dll1 even if they had been sorted as Dll1- cells. This is acknowledged by the authors ("Dll1- tumor cells gain Dll1 expression after chemoresistance) and evident in the new Fig. 6H; however, the authors claim the levels of Dll1 expression are significantly lower in the Dll1- tumours. We all know that immunofluorescence cannot quantify levels of expression and indeed, in Fig. S8d, many cells still present mCherry fluorescence, but it looks like the laser intensity has been reduced to show lower expression levels. In the rebuttal letter, the authors state "We have now added Dll1+ and Dll1- tumors stained with mCherry antibody"; however, I found no information on such an antibody in Methods or in the legends to Fig. 6 and S8, so I believe that they are showing direct mCherry fluorescence.

Another remarkable example of figure replacement comes from Figure 3D: "Our original zoomed-in

image in Fig. 3d does not show the correct proportion of K14+ cells; therefore, we have replaced Fig. 3d". The new Figure 3D shows K14+ cells that are facing the gland lumen...? Not sure if these are really basal cells? Also, in the tumour section presented in Figure 3F, a cluster of K14+ (green) cells appears on one side of the figure, but in reality, we know from the FACS gates now presented in Fig. S3 that the basal cells have almost disappeared in these luminal tumours, so it's not clear what is the identity of the K14+ cell cluster in the new Figure 3F.

One last critical remark concerns the conclusions reached by the authors on Dll1-Notch signalling, a point which was also raised by the second reviewer. The authors propose that Notch target genes would be upregulated in Dll1+ cells, completely disregarding the immense literature on Delta-Notch cis-inhibition and basing their conclusion on a simple RT-PCR experiment on mCherry+ and mCherry- cells analysed for a handful of Notch-related genes (Figure R3). This experiment, that the author chose not to include in the manuscript, presents very modest differences in expression of selected Notch-related genes and the statistics showing significance of these results are missing. Moreover, it is well established that RNA levels for Notch receptors do not necessarily reflect protein expression nor pathway activation.

Reviewer #2:

Remarks to the Author:

The manuscript does address my concerns with the original submission. It is significantly improved and deserves to be seriously considered for publication.

Reviewer #3:

Remarks to the Author:

In this revised manuscript, Kumar et al characterized the Dll1+ breast tumor cells in driving chemoresistance through the regulation of the NF-kb survival pathways. This reviewer was invited to evaluate the aspects on transcriptional and epigenetic analysis described in the revised manuscript. Thus, this reviewer deferred to the other two reviewers on other aspects of this study, and only focused on the RNA-seq and ATAC-seq studies and relevant conclusions.

Specific comments:

1. In Fig. 5, instead of only showing selective pathways, the RNA-seq and ATAC-seq analysis would benefit from more global and unbiased analysis of the altered pathways in Dll1+ vs Dll1- tumor cells. For example, what are the most significantly enriched GSEA or gene ontology pathways in the differentially expressed genes and/or ATAC-seq peaks? Is NF-kb pathway identified as the top enriched pathway from the unbiased analysis? Are the other selected pathways shown in Fig. 5a-e also within the top enriched pathways? These studies would provide stronger rationales for the focus on the NF-kb pathways as the underlying mechanism for Dll1-mediated phenotypes.

2. The RNA-seq and ATAC-seq studies would also benefit from integrative analysis to identify genes with altered gene expression (by RNA-seq) as well as differential chromatin accessibility (by ATAC-seq) in Dll1+ vs Dll1- tumor cells. These results may distinguish Notch/Dll1 direct vs indirect gene targets.

3. The conclusion that "...we didn't observe a significant difference in the genome-wide chromatin accessible regions between Dll1+ and Dll1- tumor cells (Supplementary Fig. 5f)..." is somewhat confusing. The peak distribution analysis shown in Supplementary Fig. 5f does not consider the quantitative difference in ATAC-seq signals at given genomic loci. The volcano plot in Fig. 5f indicates that there are substantial differences in ATAC-seq signals at the global scale. To make strong conclusions, the authors would need more quantitative analysis of ATAC-seq data, as well

as the integrative analysis with RNA-seq results as described above.

Reviewer #1 (Remarks to the Author):

The revised manuscript has corrected some of the problems existing in the original work, but unfortunately the authors have failed to address several major concerns that were raised by reviewers. Suggested experiments to improve or clarify the results were in most cases not performed or disregarded and the authors insist on drawing strong conclusions that, in my view, are not supported by solid experimental evidence.

We respectfully disagree with the reviewer that we did not follow his/her suggestions in the previous revision. We performed several experiments and made necessary changes to improve the manuscript in the earlier revision. In the current revision, we have further strengthened the molecular mechanism by adding integrated RNA-seq and ATAC-seq data. We have also removed Figure 3 with lineage tracing as we don't have sufficient data to make that claim strongly. Furthermore, we have also added additional basal marker K5, to show the distribution of basal cells in these tumors in the context of Dll1 (please see Supplementary Fig. 2a-b).

The most critical concern in my opinion is related to lineage tracing analysis presented in Figure 3, which I advise the authors to remove from the study, as it is misinterpreted and will generate unnecessary confusion. Indeed, it is not just the term "cell fate change" that I was criticising, but the idea that Dll1+ basal cells would give rise to luminal cells within tumours. Clearly PyMT tumours, unlike the MMTV-Wnt1 tumours (see also Fig. R1), present a major shift in the basal to luminal ratio (now shown in Fig. S3), which is reflected by the expansion of the luminal cells that express Dll1. This does NOT mean that these cells derive from basal cells and in fact in the new FACS gating strategy in Fig. R2, one can appreciate that many Tomato+ cells are found in the stromal fraction (29.1%!). This does not mean that these stromal cells derive from Dll1+ basal cells, so why should the authors conclude that it is the case for the increased Dll1+ luminal cell fraction?

We agree with the reviewer that without appropriate experiments, we are unable to trace the lineage of luminal Dll1+ tumor cells carefully. Hence, we have removed Figure 3 from the revised manuscript.

In the legend to Figure 2 we read: "Fig. 2: Dll1+ basal cells change to K8+ luminal cells during PyMT tumor development". This is not correct. In this Figure, they use a reporter mouse that follows Dll1 expression, they DO NOT trace Dll1-expressing cells. The data simply showed show the shift of Dll1 expression from basal cells in physiological conditions to luminal cells K8+ in tumors.

Following the reviewer's suggestion, we have modified the sentence to "Fig. 2: Number of Dll1+ luminal cells increases during PyMT tumor development" on page number 24.

In the rebuttal the authors claim that "The highlight of the manuscript is determining the molecular mechanism of Dll1 on tumor cells survival after chemoresistance by NF- κ B signaling, which is where we focused for the revision"; this is the main argument to explain why they avoided to address my question regarding whether luminal Dll1-expressing cells are, indeed, the cells of origin of luminal tumors with lack of evidence for a cell fate change. If that is their final argument, it is even more justifiable to remove the entire Figure 3 and clarify the message.

IF for K5, p63 or SMA, as I had requested, is absolutely required to explore the possibility of double positive cells and to explain the "disappearance" of K14+ cells in PyMT tumours.

Moreover, in Figure 3, an increase from 11 to 14.9% in the percentage of Tom+ cells at hyperplasia stage and advanced tumor stage is reported; however, the images are not representative of such percentages.

Please see earlier comment about removing Figure 3 lineage tracing part.

We have now provided the double staining images of K5 (new marker added in this revision), K8 markers along with mCherry using hyperplasia and tumor sections (Supplementary Fig. 2a-b). We found some co-staining for some Dll1+ cells and basal K5 marker in hyperplasia and tumor sections, corroborating with the RNA-seq data.

However, Dll1⁺ co-staining with K8 was very prominent, especially in the tumor (Rebuttal figure 1a-b). In addition to K14, we have also added a new K14 antibody staining which corroborates earlier K14 data in the manuscript. This new K14 images are in Rebuttal Figure 2a-b and can be added as Supplemental figures if need be. These data clearly show the localization of basal cells in the tumors. Also, we have added zoomed out images of tumors to show a broader view of the field to demonstrate the overall distribution of all these proteins in the tumors (Rebuttal Figure 2a).

As we have removed the figure 3, therefore, no changes were made on representative image replacement Tomato⁺ cells in figure 3.

In several instances, I found it troubling that the authors acknowledged my remarks and decided to replace specific images with other ones that better agree with the conclusions they have reached. How do we know which images are representative of the results, the originally submitted ones or the revised ones?

First, we want to thank the reviewer for pointing out problematic representative images in the previous round of revision. We apologize for not clarifying the selection of representative images. The rationale for choosing correct representative images was that we quantified many images to choose one which represents the real scenario closest. Hence, we are confident that the current images are true representative images. We have provided additional representative images (3 extra panels besides the one in manuscript to show representation of the image) for each image for figs 2d, 2e, 3f, 5e, 5g-h, 6f, supp. Figs. 2d, 2f, 4a-d, 8c-d. Please see figures in Supplementary Fig.11 (new representative images).

One striking example relates to my observation that cells surviving chemotherapy start expressing Dll1 even if they had been sorted as Dll1⁻ cells. This is acknowledged by the authors (“Dll1⁻ tumor cells gain Dll1 expression after chemoresistance) and evident in the new Fig. 6H; however, the authors claim the levels of Dll1 expression are significantly lower in the Dll1⁻ tumours. We all know that immunofluorescence cannot quantify levels of expression and indeed, in Fig. S8d, many cells still present mCherry fluorescence, but it looks like the laser intensity has been reduced to show lower expression levels. In the rebuttal letter, the authors state “We have now added Dll1⁺ and Dll1⁻ tumors stained with mCherry antibody”; however, I found no information on such an antibody in Methods or in the legends to Fig. 6 and S8, so I believe that they are showing direct mCherry fluorescence.

We thank the reviewer for understating correctly that cells surviving chemotherapy start expressing or enhance Dll1 levels. We regret for not writing the details of the experiment. In old figures 6h (New Figure 5h) and S8d, we performed IF on formalin-fixed tumors with mCherry antibody to detect Dll1-expressing cells. We showed the abundance of Dll1⁺ cells in Dll1⁻ and Dll1⁺ tumors. The mCherry antibody information was available in the supplemental Table 2. Now, we have added the experiment details in the figure legend to make it clearer. Also, please see the representative images (3 extra panels per figure and also low magnification images to show the larger area, see Supplementary Fig. 11). From these images it is obvious that although Dll1⁻ tumors gain Dll1⁺ mCherry expression after chemoresistance, that expression is quite low compared to Dll1⁺ cells in tumor, which increases their Dll1⁺ expression much more.

Another remarkable example of figure replacement comes from Figure 3D: “Our original zoomed-in image in Fig. 3d does not show the correct proportion of K14⁺ cells; therefore, we have replaced Fig. 3d”. The new Figure 3D shows K14⁺ cells that are facing the gland lumen...? Not sure if these are really basal cells? Also, in the tumour section presented in Figure 3F, a cluster of K14⁺ (green) cells appears on one side of the figure, but in reality, we know from the FACS gates now presented in Fig. S3 that the basal cells have almost disappeared in these luminal tumours, so it's not clear what is the identity of the K14⁺ cell cluster in the new Figure 3F.

Please see our response above to choose a representative image based on a quantification of multiple images. Also please see supplemental figures showing the additional images (Supplementary Fig. 11). Some Dll1⁺ cells colocalize with K14/K5 and some do not suggesting basal cell heterogeneity in these tumors (Supplementary

Fig. 2a-b and Fig. 2d-e).

One last critical remark concerns the conclusions reached by the authors on Dll1-Notch signalling, a point which was also raised by the second reviewer. The authors propose that Notch target genes would be upregulated in Dll1⁺ cells, completely disregarding the immense literature on Delta-Notch cis-inhibition and basing their conclusion on a simple RT-PCR experiment on mCherry⁺ and mCherry⁻ cells analysed for a handful of Notch-related genes (Figure R3). This experiment, that the author chose not to include in the manuscript, presents very modest differences in expression of selected Notch-related genes and the statistics showing significance of these results are missing. Moreover, it is well established that RNA levels for Notch receptors do not necessarily reflect protein expression nor pathway activation.

We would like to highlight that Notch target genes are upregulated in Dll1⁺ tumor cells is also supported by RNA-seq data in supplementary figure 5e. Core genes highlight several Notch target genes (Dll1, Prkca, Ppard, Hes1, Notch3, Notch1, etc.). Western blot to show protein data has a technical issue as millions of cells are required from sorting which will require many tumors. Therefore, we have incorporated a paragraph in the discussion stating that we need further evaluations to understand the Notch signalling, which happens between Dll1⁺ cells and Notch receptor-expressing cells. Please see page 14, last paragraph.

Reviewer #2 (Remarks to the Author):

The manuscript does address my concerns with the original submission. It is significantly improved and deserves to be seriously considered for publication.

We are very thankful to the reviewer for his/her time and helpful comments. We appreciate him/her for recommending the manuscript for publication.

Reviewer #3 (Remarks to the Author):

In this revised manuscript, Kumar et al characterized the Dll1⁺ breast tumor cells in driving chemoresistance through the regulation of the NF- κ B survival pathways. This reviewer was invited to evaluate the aspects on transcriptional and epigenetic analysis described in the revised manuscript. Thus, this reviewer deferred to the other two reviewers on other aspects of this study, and only focused on the RNA-seq and ATAC-seq studies and relevant conclusions.

We are thankful to the Editor for inviting her/him to review the bioinformatics data.

Specific comments:

1. In Fig. 5, instead of only showing selective pathways, the RNA-seq and ATAC-seq analysis would benefit from more global and unbiased analysis of the altered pathways in Dll1⁺ vs Dll1⁻ tumor cells. For example, what are the most significantly enriched GSEA or gene ontology pathways in the differentially expressed genes and/or ATAC-seq peaks? Is NF- κ B pathway identified as the top enriched pathway from the unbiased analysis? Are the other selected pathways shown in Fig. 5a-e also within the top enriched pathways? These studies would provide stronger rationales for the focus on the NF- κ B pathways as the underlying mechanism for Dll1-mediated phenotypes.

This is a great suggestion by the reviewer. In RNA-seq, the NF- κ B pathway was a top altered pathway in both mCherry⁺ (Dll1⁺ cells) (NES=1.74 at 1st rank among hallmark signatures) and GFP⁺ (Dll1⁺ cells) (NES=2.32 at 3rd rank among hallmark signatures) at very significant FDR values. Note: Two Dll1 reporter models were

used for RNA-seq analyses. This enrichment of NF- κ B pathway in Dll1⁺ cells were also observed in other datasets of GSEA such as C2 (GFP reporter, TIAN_TNF_SIGNALING_VIA_NFKB, NES=2.02, REACTOME_TRAF6_MEDIATED_NF_KB_ACTIVATION, NES=1.68).

As per suggestion of the reviewer, we performed an integrated analysis of RNA-seq and ATAC-seq data and found that, on a global level, there is overlap of genes between the two analyses. This overlap is more in the upregulated/increased accessibility genes, rather than the downregulated/decreased accessibility groups. For the integrated analysis, we took the "ATAC-seq Up" signature (all genes with a proximal ATAC-seq peak increasing significantly at p-adjusted < 0.05) and tested it for enrichment in the GFP/mCherry Up combined RNA-seq ranked list. We found that this ATAC-seq signature is strongly enriched in the RNA-seq data (NES = 2.24, p<0.001). Please see new figure in manuscript (Figure 4e). In contrast, the "ATAC-seq Down" signature (genes with decreased chromatin accessibility in Dll1⁺ cells) showed a trend toward global transcriptional downregulation in Dll1⁺ cells, but fell short of statistical significance (NES = -1.23, p = -0.145) and was thus not included in the manuscript.

We focused further analyses on the 23 GSEA "Core Enrichment" genes that are both (1) in the ATAC-seq Up signature, and (2) transcriptionally upregulated in Dll1⁺ cells (Rebuttal figure 3a-c, next page). Please see figure in revised manuscript figure 4e. Enrichr analysis of these 23 common upregulated/increased accessibility genes from Dll1⁺ cells shows significant enrichment of both Notch signaling and NF- κ B signaling (RelA). Please see Rebuttal figures 3a-c below and Suppl Fig. 5h. Thus, our claim on choosing NF- κ B pathway downstream of Dll1-Notch signaling for tumor cells survival and resistance phenotype is further strengthened. Indeed Dll1⁺ tumors show increased nuclear translocation of NF- κ B (RelA) protein compared to Dll1⁻ tumors (Figure 4h) supporting the ATAC-seq and RNA-seq data.

a Enriched in ATAC-seq/RNA-seq GSEA enrichment core (integrative)

TRANSFAC and JASPAR PWMs

Bar Graph Table Grid Network Clustergram

Click the bars to sort. Now sorted by p-value ranking.

b Enriched in ATAC-seq/RNA-seq GSEA enrichment core (integrative)

BioPlanet 2019

Bar Graph Table Clustergram

Click the bars to sort. Now sorted by p-value ranking.

c Enriched in ATAC-seq/RNA-seq GSEA enrichment core (integrative)

CORUM

Bar Graph Table Grid Network Clustergram

Click the bars to sort. Now sorted by p-value ranking.

Rebuttal Figure 3a-c. Gene Ontology enrichment analysis for differentially up-regulated genes in Dll1⁺ cells compared with Dll1⁻ cells after ATAC-seq and RNA-seq integrated analyses.

We also investigated pathway-level changes in ATAC-seq data independently from RNA-seq trends, to make sure that our integrative ATAC/RNA-seq analysis did not miss any notable trends that may be present in ATAC-seq but not in RNA-seq data. We again found significant indication of alterations in several Notch pathway and NF- κ B signaling (Rebuttal figure 4a-f). In RNA-seq data, NF- κ B signalling was either the first one or among the top 10 signatures to be different between Dll1⁺ and Dll1⁻ cells.-

a Enriched in ATAC-seq in Dll1-high cells (without considering RNA-seq)

<https://tripod.nih.gov/bioplanet/>

BioPlanet 2019

Bar Graph Table Clustergram

Click the bars to sort. Now sorted by p-value ranking.

b Enriched in ATAC-seq in Dll1-high cells (without considering RNA-seq)

Reactome 2016

Bar Graph Table Clustergram

Click the bars to sort. Now sorted by p-value ranking.

c Enriched in ATAC-seq in Dll1-high cells (without considering RNA-seq)

NCI-Nature 2016

Bar Graph Table Clustergram

Click the bars to sort. Now sorted by p-value ranking.

d Enriched in ATAC-seq in Dll1-high cells (without considering RNA-seq)

<https://bioplex.hms.harvard.edu/>

e Enriched in ATAC-seq in Dll1-high cells (without considering RNA-seq)

<http://mips.helmholtz-muenchen.de/corum/>

f Enriched in ATAC-seq in Dll1-high cells (without considering RNA-seq)

Rebuttal Figure 4a-f. Gene Ontology enrichment analysis for differentially up-regulated genes in Dll1⁺ cells compared with Dll1⁻ cells after ATAC-seq analysis.

For figures 5a-e, we found several signatures to be enriched in Dll1⁺ tumor cells in RNA seq. However, from ATAC-seq analysis, Notch signaling, hypoxia signalling and NF-κβ signaling were the ones among those 5 signatures to be significantly enriched in the Dll1⁺ tumor cells. This could be due to the fact that ATAC-seq was performed with only one reporter model (Dll1^{mecherry}), whereas in RNA-seq, Dll1⁺ cells were obtained from both

Dll1^{mCherry} and Dll1^{GFP} models. It is possible that addition of another reporter model in ATAC-seq may make the comparison between ATAC-seq and RNA-seq more comprehensive. Notably, Notch and NF- κ B signaling is the common ones which is the focus of the manuscript.

2. The RNA-seq and ATAC-seq studies would also benefit from integrative analysis to identify genes with altered gene expression (by RNA-seq) as well as differential chromatin accessibility (by ATAC-seq) in Dll1⁺ vs Dll1⁻ tumor cells. These results may distinguish Notch/Dll1 direct vs indirect gene targets.

Please see the detailed response to comment 1 above. We did perform integrated analysis of RNA-seq and ATAC-seq and found that 23 genes were upregulated in both RNA-seq and ATAC-seq. These genes include Notch signaling targets (Dll1, Efnb2) and also contains RelA (part of the NF- κ B transcription factor heterodimer). Further examination of these genes by EnrichR shows that Notch signaling and NF- κ B signaling are part of the common pathways, suggesting a strong connection of Dll1 in Notch signaling and NF- κ B signaling. Please see Rebuttal Figure 3a-c and Fig. 4e and Suppl Fig. 5h.

3. The conclusion that “...we didn’t observe a significant difference in the genome-wide chromatin accessible regions between Dll1⁺ and Dll1⁻ tumor cells (Supplementary Fig. 5f)...” is somewhat confusing. The peak distribution analysis shown in Supplementary Fig. 5f does not consider the quantitative difference in ATAC-seq signals at given genomic loci. The volcano plot in Fig. 5f indicates that there are substantial differences in ATAC-seq signals at the global scale. To make strong conclusions, the authors would need more quantitative analysis of ATAC-seq data, as well as the integrative analysis with RNA-seq results as described above.

We apologize for confusion over figure S5f (now figure S5g). What we had intended to convey is that, on a very global level, ATAC-seq data do not indicate any dramatic, epigenome-wide changes in chromatin accessibility patterns between Dll1⁺ and Dll1⁻ cells. This conclusion is based on finding similar numbers of UTR, exon, intergenic, intron, and promoter peaks in both Dll1⁺ and Dll1⁻ tumor cells, as shown in figure S5g. This finding is rather expected, given that these cells are of the same lineage, and as Dll1 is not itself a chromatin-modifying enzyme. Thus, we focused our attention instead on specific genes with differential chromatin accessibility in Dll1⁺ vs. Dll1⁻ cells. As shown in the volcano plot in Fig 4f, Dll1⁺ cells have increased chromatin accessibility at many important genes such as Dll1, NF- κ B1 and other metastasis related genes. Please also see above response to comments on more integrated approach of RNA-seq and ATAC-seq.

Reviewers' Comments:

Reviewer #1:

Remarks to the Author:

In this new round of revision, I appreciate that the authors have removed Figure 3, although they did not exclude the possibility of a cell fate change in the Discussion (p.14), see my comment below. I also asked for consistency in the representative pictures selected and they indeed provided extra figures to demonstrate that their images are representative of several experiments. Still, Figure 2 and all images in supplemental Fig. S11 show a vast majority of luminal Dll1-positive cells, whereas the quantification by flow cytometry reports 58.8% basal cells vs 24.2% luminal cells within the Dll1-positive gate in hyperplasia stage (Supplemental Fig. S3b). I do not yet understand the contradiction in these data.

Moreover, notwithstanding what they state in the rebuttal letter, when I read the revised text I could not appreciate many changes and the conclusions that the authors reach remain the same as in the initial submission. In the Discussion the authors:

- did not change the conclusion of a possible cell fate change of Dll1+ cells, which was removed from the results for lack of evidence (old Fig. 3): "...could be due to a cell fate change of Dll1 from basal to luminal fate during tumor progression" (p.14)
- did not change their conclusion on Notch target activation in the Dll1+ signal-sending cells: "The RNA-seq data indicated that Notch target genes are activated in Dll1+ tumor cells compared with Dll1- tumor cells".

Incidentally, the genes encoding Delta, Notch1 and Notch3 are not "Notch target genes", but rather they belong to the Gene Set: HALLMARK_NOTCH_SIGNALING, which is not the same.

Furthermore, the new figures provided in the rebuttal as well as in Supplementary Fig. 11 are of extremely poor resolution, they are very pixelated and they present very low magnifications, making it impossible to appreciate double positive cells Dll1+/K14+, as claimed by the authors.

Even the typos from the previous version are not corrected: i.e. see "in this tumors" at page 13 and "tumor progrersses" at page 14.

Minor comments:

It's not NF κ B. This pathway is called NF κ B (Nuclear Factor Kappa B).

MMTV-PyMT tumours are a model for luminal breast cancer and not basal.

Reviewer #3:

Remarks to the Author:

The authors have adequately addressed my questions on the integrative analysis of RNA-seq and ATAC-seq datasets by additional data analysis and/or clarifications. The new results included in the revised manuscript and the rebuttal letter support the conclusions that the activation of NF- κ B pathways is downstream of Dll1 and associates with the chemoresistant phenotype. This reviewer has no additional questions and is supportive of the publication of this important study.

Reviewer #1 (Remarks to the Author):

In this new round of revision, I appreciate that the authors have removed Figure 3, although they did not exclude the possibility of a cell fate change in the Discussion (p.14), see my comment below. I also asked for consistency in the representative pictures selected and they indeed provided extra figures to demonstrate that their images are representative of several experiments. Still, Figure 2 and all images in supplemental Fig. S11 show a vast majority of luminal Dll1-positive cells, whereas the quantification by flow cytometry reports 58.8% basal cells vs 24.2% luminal cells within the Dll1-positive gate in hyperplasia stage (Supplemental Fig. S3b). I do not yet understand the contradiction in these data.

We have now removed the text with a possibility of cell fate change in discussion, p14. We appreciate the reviewer's positive comments on the figures. The reviewer has raised an interesting point. The difference we observed in FACS vs IF could be because there is lot of heterogeneity in structure of hyperplasia in MMTV-PyMT model during transition to hyperplasia stage from a normal mammary gland state. In normal mammary gland, Dll1⁺ cells (mcherry⁺ or GFP⁺) are predominantly in basal (Chakrabarti et al, Science, 2018) and during transition to hyperplasia, the Dll1⁺ cells increase in the luminal compartment. Thus, it is possible that in figure Fig. S3b, the mammary glands are in early hyperplasia stage with still several areas of normal mammary gland which typically has more Dll1⁺ cells in basal area. Alternatively, it is also possible that some Dll1⁺ basal cells mark additional basal genes besides K14, which are part of CD24⁺CD29^{high} population seen in FACS.

Moreover, notwithstanding what they state in the rebuttal letter, when I read the revised text I could not appreciate many changes and the conclusions that the authors reach remain the same as in the initial submission. In the Discussion the authors:

- did not change the conclusion of a possible cell fate change of Dll1⁺ cells, which was removed from the results for lack of evidence (old Fig. 3): "...could be due to a cell fate change of Dll1 from basal to luminal fate during tumor progression" (p.14)

We understand the reviewer's concern and following his/her suggestion, we have removed the whole sentence from the revised manuscript.

- did not change their conclusion on Notch target activation in the Dll1⁺ signal-sending cells: "The RNA-seq data indicated that Notch target genes are activated in Dll1⁺ tumor cells compared with Dll1⁻ tumor cells". Incidentally, the genes encoding Delta, Notch1 and Notch3 are not "Notch target genes", but rather they belong to the Gene Set: HALLMARK_NOTCH_SIGNALING, which is not the same.

We have modified the sentence to "The RNA-seq data indicated that Notch signalling genes are up in Dll1⁺ tumor cells compared with Dll1⁻ tumor cells".

Furthermore, the new figures provided in the rebuttal as well as in Supplementary Fig. 11 are of extremely poor resolution, they are very pixelated and they present very low magnifications, making it impossible to appreciate double positive cells Dll1⁺/K14⁺, as claimed by the authors.

The images could be pixelated due to file format changing to PDF. We have provided high-quality TIFF images for the Rebuttal figures. We have also increased the size of images and added more arrows indicating double-positive cells.

Even the typos from the previous version are not corrected: i.e. see "in this tumors" at page 13 and "tumor progersses" at page 14.

Minor comments:

It's not NFκβ. This pathway is called NFκB (Nuclear Factor Kappa B).

Thank you for finding the errors. We have corrected them in the revised version.

MMTV-PyMT tumours are a model for luminal breast cancer and not basal.

To our best of knowledge, we did not mention in the manuscript that “MMTV-PyMT is basal”. So, we are not sure of the reviewer’s comment.

We thank the reviewer for his/her thorough review and comments to make the manuscript better for publication.

Reviewer #3 (Remarks to the Author):

The authors have adequately addressed my questions on the integrative analysis of RNA-seq and ATAC-seq datasets by additional data analysis and/or clarifications. The new results included in the revised manuscript and the rebuttal letter support the conclusions that the activation of NF- κ B pathways is downstream of Dll1 and associates with the chemoresistant phenotype. This reviewer has no additional questions and is supportive of the publication of this important study.

We are very thankful to the reviewer for his/her critical inputs regarding RNA-seq and ATAC-seq data. We appreciate your helpful comments.

b, cropped

b, uncropped